# Embroidering the Life of Thomas Becket during the Middle Ages: Cult and Devotion in Liturgical Vestments

## Nathalie Le Luel

Faculty of Humanities, Université Catholique de l'Ouest, 3 Pl. André Leroy, B.P. 10808, 49008 Angers, France; nathalie.leluel@uco.fr

**Abstract:** From the early studies of Tancred Borenius (1885–1948) to the present, the iconography of the archbishop Thomas Becket has drawn attention among scholars. Numerous studies have been published on the representation of Becket's martyrdom in mural painting, sculpture, and reliquary caskets. Despite this attention, many questions concerning the selection of episodes embroidered in liturgical vestments and textiles, as well as the commissioning of these objects, remain unresolved. How devotion to Becket spread globally in the Western world has not yet been satisfactorily determined, and there may have been a number of different factors and transmitters. Thus, medieval embroidery could also have been a driving force behind the development and the dissemination of Becket's cult—notably in the ecclesiastical and, more specifically, episcopal milieu across the Latin Church. This type of production quickly reached ecclesiastical patrons, who were interested in the opportunity of wearing a headpiece or vestments (copes and chasubles) that would serve as reminders of the Archbishop of Canterbury. This was the perfect opportunity for a papal curia that, since Alexander III, had tasked itself with promoting Thomas Becket's legacy, integrating the saint within Christian martyrial history and within a history of a militant Church.

**Keywords:** medieval embroidery; *Opus Anglicanum*; textile creation; liturgical vestments; Becket's iconography; saint's devotion and cult; ecclesiastical patrons

## 1. Introduction

In 1538, in the course of the Henrician Reformation, King Henry VIII (1509–1547) ordered the systematic destruction of all relics of Thomas Becket and the eradication of his cult. The king's instruction, emblematic in its extreme violence, recalled the troubled relationship which had existed between the English monarchy and the Archbishops of Canterbury since the 12th century, and symbolically replayed Becket's brutal death, assassinated in his cathedral on 29 December 1170 at the culmination of a long battle with Henry II (Henry Plantagenet)[1]. The consequences of Henry VIII's decision escalated during the regency of Edward VI, Henry VIII's son, with injunctions to destroy all 'superstitious images' imposed in 1547 (Aston 2016). Thus, the majority of sculpted and painted decorations devoted to Thomas Becket, as well as a great variety of furnishings (manuscripts, vestments, fabrics, liturgical furniture, medals, etc.) on which he appeared, were destroyed or irreversibly desecrated.

Among these objects, a number of fabric pieces were either burned or the image of Thomas Becket was removed. In addition to these losses, caused by the wave of iconoclasm that swept across 16th century England, many textiles and items of clothing were lost during the Middle Ages. Notwithstanding the disproportionately reduced body of evidence, some sacerdotal vestments and liturgical items featuring or representing the Archbishop of Canterbury are still conserved[2]. Preserved in collections dispersed across Europe, they were almost all produced in England (*Opus Anglicanum*[3]) but were, for the most part, transported to the continent before the Reformation[4]. Some of these pieces were produced

only shortly after the saint's canonisation in 1173, whilst others are proof of a cult that was still active at the end of the Middle Ages.

Thomas Becket was the subject of a great deal of iconography between the 12th and 15th centuries, as many studies have already shown[5]. This new analysis highlights how the textile domain, particularly the art of embroidery, is evidence for the development and diffusion of the cult of Becket—notably in the ecclesiastical and, more specifically, episcopal milieu across the Latin Church. On what types of vestments (mitres, copes, chasubles, etc.) do images of the saint appear? Which images take precedence? These images can also be associated with other scenes from the biblical or hagiographical repertoire. Moreover, we pay attention to the ecclesiastical patrons whose devotion to Thomas Becket led them to visually translate his image into the clothes they wore[6].

Excluding textual references, today we conserve 18 fabric pieces which show the English saint as either an isolated image or within scenes of his life. The majority of the embroidered items (16) take the form of liturgical vestments, including four mitres, seven copes (or fragments of copes), one chasuble, two fragments of an orphrey from a cope or chasuble, and two maniples. The remaining two examples are liturgical textiles, including fragments of what was probably an altar frontal or a reliquary sheet, and lastly a pall[7]. The chronological breadth of the collection spans four centuries, from the end of the 12th century to ca. 1500, with creative activity concentrated in the 13th and 14th centuries, at a time when *Opus Anglicanum* reached a technical apogee. This period also witnessed the creation of monumental decorations devoted to the saint, as well as a considerable proportion of the iconographic production, all fields included, dedicated to Thomas Becket.

## 2. Mitres

The four mitres in the corpus form the oldest group of items and show that Becket was represented in the field of liturgical vestments very soon after his death. The headpiece, worn during liturgical ceremonies and, since the beginning of the Church, mostly reserved for bishops, is an object of eminent symbolism, and we note that this first textile iconography of the saint appears on this fundamental piece of the episcopal wardrobe. Produced at the turn of the 12th–13th centuries (1180–1250) and embroidered on silk (samite), the four mitres of interest were all produced in England but, at dates unknown, were moved to the continent. They serve as some of the oldest evidence of the significant serial production which spread the cult of the archbishop, canonised not long previously (Geijger 1956; Vogt 2010; Coatsworth and Owen Crocker 2018, pp. 29–31, 62–65). One of the mitres is conserved in Namur (Belgium), as part of the Treasure of Oignies (Figures 1 and 2)[8]. The mitre is thought to have belonged to Jacques de Vitry, bishop of Acre, in the Holy Land (1216–1240).

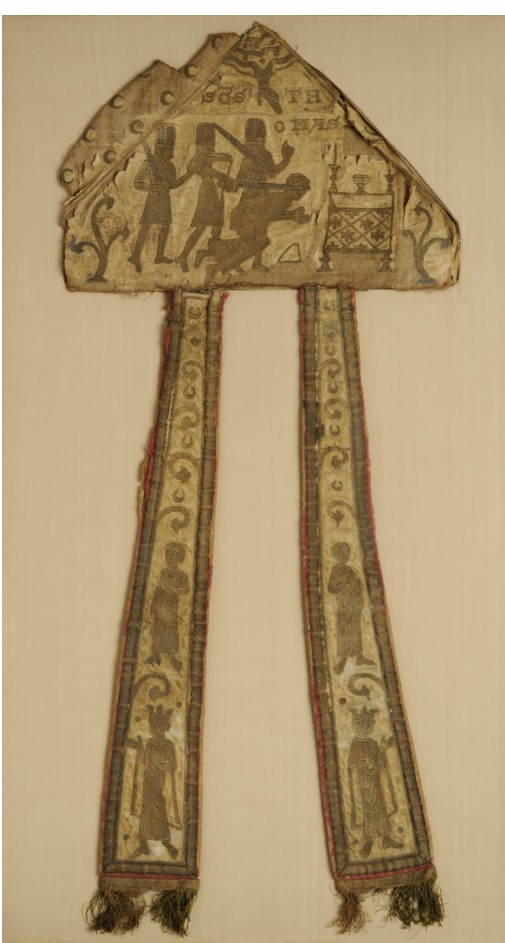

**Figure 1.** Jacques de Vitry's Mitre, front face, early 13th century. Donated by the Sisters of Notre-Dame de Namur, coll. King Baudouin Foundation, on deposit at the Société Archéologique of Namur and exhibited at the TreM.a—Musée des Arts Anciens, Namur, Belgium © Hughes Dubois.

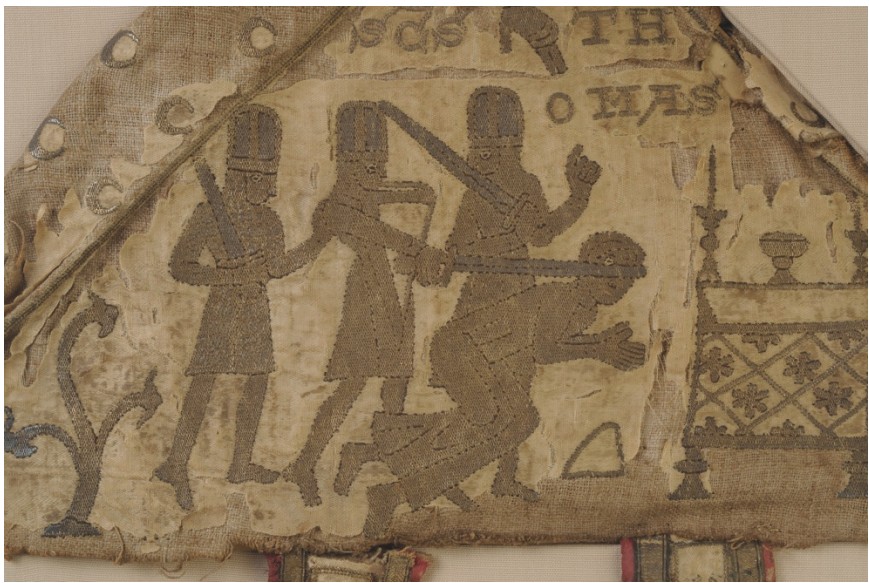

**Figure 2.** Jacques de Vitry's Mitre, detail of the front face: Thomas Becket murder. Donated by the Sisters of Notre-Dame de Namur, coll. King Baudouin Foundation, on deposit at the Société Archéologique of Namur and exhibited at the TreM.a—Musée des Arts Anciens, Namur, Belgium © Hughes Dubois.



The mitre that is preserved at the Bayerisches Nationalmuseum in Munich (Germany) is believed to have been offered to the Cistercian Abbey of Seligenthal in Landshut (Bavaria), by the Duchess Ludmilla (ca. 1170–1240), founder of the Seligenthal convent (Figures 3 and 4)[9].

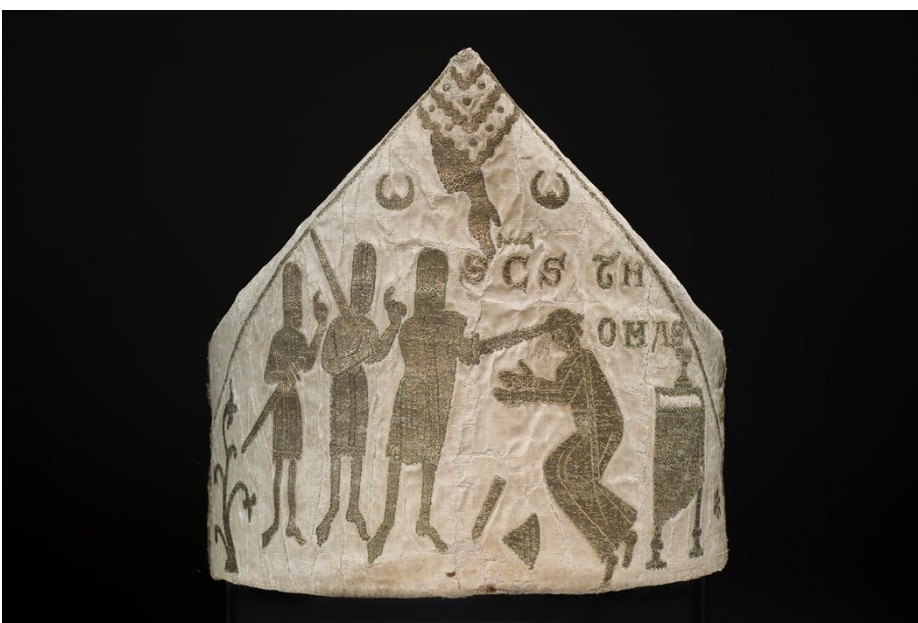

**Figure 3.** Seligenthal Mitre, front face with the martyrdom of Becket, early 13th century. Munich, Bayerisches Nationalmuseum, Germany. © Munich, Bayerisches Nationalmuseum.

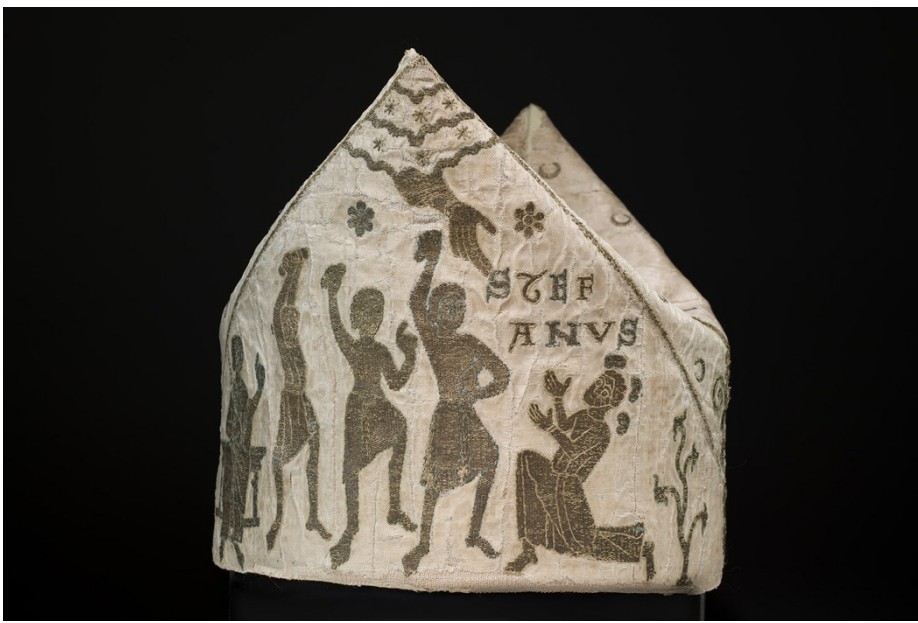

**Figure 4.** Seligenthal Mitre. rear face with the Stoning of St Stephen. Munich, Bayerisches Nationalmuseum, Germany. © Munich, Bayerisches Nationalmuseum.

The third mitre comes from the treasury of Sens cathedral (Figures 5 and 6) where Becket sought refuge during his exile to France (Becket was a guest at the abbey Sainte-Colombe at Saint-Denis-lès-Sens between 1166 and 1170) and where the cult of the English saint developed early[10].

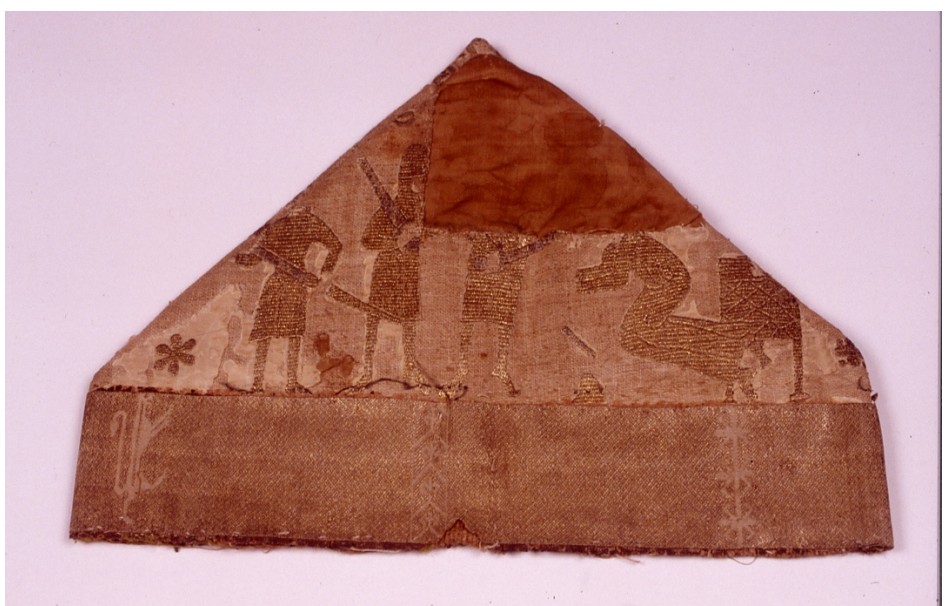

**Figure 5.** Sens Mitre, front face with the martyrdom of Becket, early 13th century. Musées de Sens, France. © Musées de Sens—E. Berry.

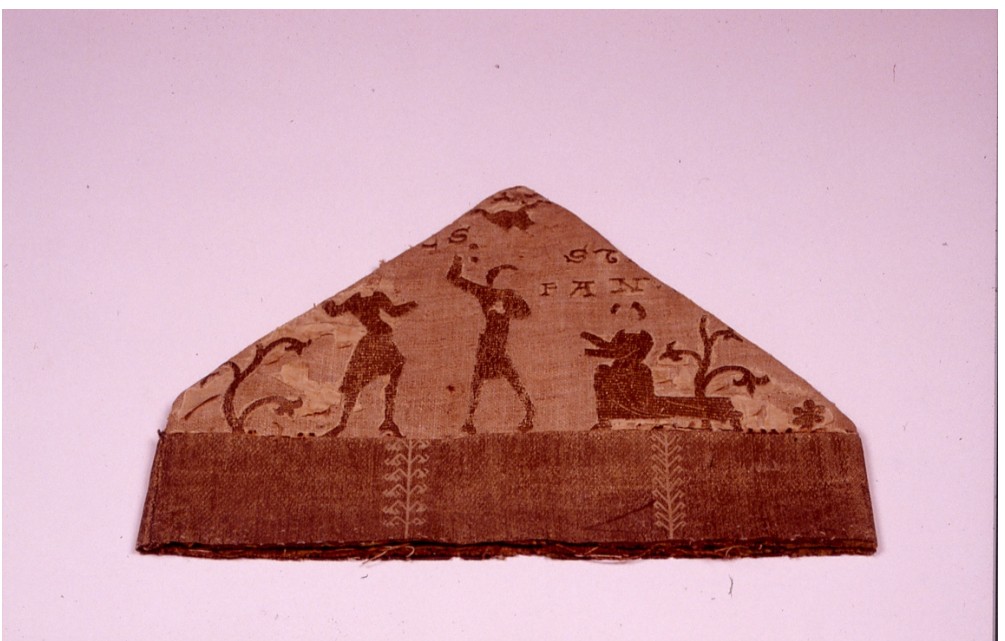

**Figure 6.** Sens Mitre, rear face with the Stoning of St Stephen. Musées de Sens, France. © Musées de Sens—E. Berry.

Lastly, the fourth mitre is located in Italy (Figures 7 and 8), where unsurprisingly it can be found in the treasury of Anagni Cathedral[11]. Anagni had been the seat of the Roman Curia and is the location of the earliest evidence pointing to a cult dedicated to Thomas Becket in the West, diffused as early as the end of the 12th century and driven by Pope Alexander III (1159–1181) who canonised Thomas Becket in 1173.

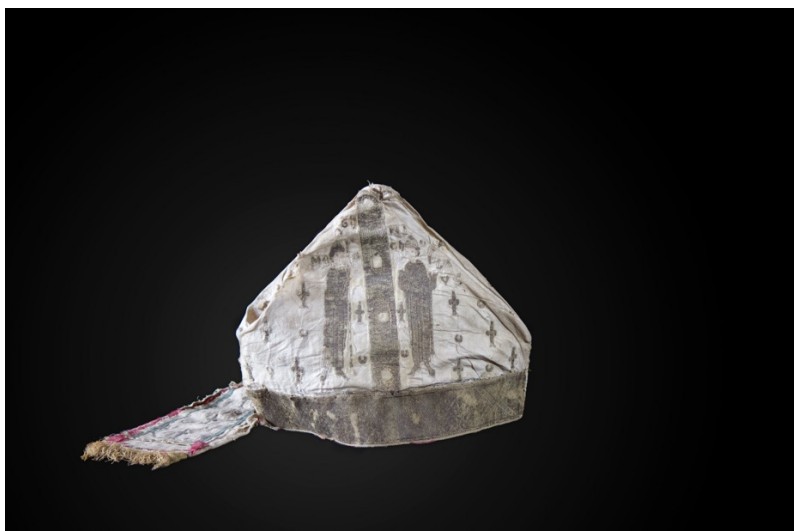

**Figure 7.** Anagni Mitre, front face with full-length portraits of Becket and St Nicholas. Ancient treasury of Anagni Cathedral, now in the cathedral museum, Italy. © by concession of the Chapter of the Basilica Cathedral of Anagni. Photo: Graframan.com.

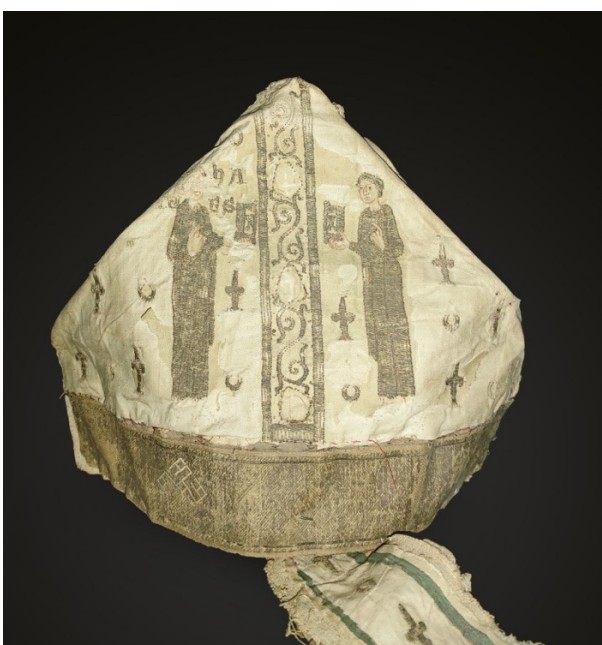

**Figure 8.** Anagni Mitre, front face with full-length portraits of Becket and St Nicholas. Ancient treasury of Anagni Cathedral, now in the cathedral museum, Italy. © by concession of the Chapter of the Basilica Cathedral of Anagni. Photo: Graframan.com.

On three of the mitres, the martyrdom of Thomas Becket appears on the forward face of the mitre, with the scene adapted to the item's triangular shape (Figures 2, 3 and 5). On the whole, the images conform to the same iconographical schema: a somewhat simplified representation of the saint's assassination, much like on other objects, such as Limoges enamel reliquary caskets, where the limited space requires a reduction in the number of protagonists. Here, Becket's assailants, thought to be four in total, are reduced to three individuals: probably comprising William of Tracy, Reginald Fitzurse, and Richard Brito. The priest Edward Grim is also missing, who is almost always present in historical images of the martyr (e.g., in BL, Harley MS 5102: Browne, Davies, Michael (ed), 2016, pp. 119–20). These gaps aside, this illustration reproduces the essential elements of the martyr's iconography, which began with illuminated manuscripts at the beginning of

the 12th century (featuring, among other objects, the altar, chalice, the fallen headpiece, and Richard Brito's broken sword). Here, we see a reduced formula of the martyrdom of Thomas Becket which, whilst the position of the saint varies from mitre to mitre, retains standardised with stylised characteristics. These mitres are examples of invention and reinvention based on an archetypal image of the murder, in much the same way as the scene is represented on the reliquary caskets that were dedicated to the saint during the same period. The mitres, as with a majority of the images on other materials featuring Becket's martyrdom, do not depart from the convention of showing Becket in front of an altar prepared for the celebration of the Mass, despite this setting being different to what historical sources suggest. In her 2004 study, Kay Slocum analysed this iconographical addition to historical reality as a symbolic construction with two objectives (Slocum 2004, p. 112): firstly, creating an opposition between the 'bloody weapons'—the swords—of Henry II's knights with the 'weapons of peace' of the archbishop who is presented as God's servant, and then creating a parallel, via the chalice evoking the Eucharist, between Becket's sacrifice and the sacrifice of Christ, a comparison that will prove significant again when we come to look at the embroidered copes in the corpus.

Becket's murder appears on the main face of the three mitres described above. In the case of the mitres conserved in Munich and at Sens Cathedral, the rear face of the item shows the Stoning of St Stephen (Figures 4 and 6), whereas the mitre in Namur shows the torture of St Lawrence, burned on a flaming grill. By contrast, the mitre conserved in Anagni shows full-length portraits of Becket and St Nicholas on the front face (Figure 7), while St John and an unidentified saint appear on the rear face (Figure 8)[12]. In this way, Thomas Becket is associated with two Christian martyrs from the outset, as well as with the Bishop of Myra[13]. Indeed, accounts of Becket tell us how he compared himself to the first Christian martyr, St Stephen. This iconographic drawing of parallels between Thomas and Stephen has been interpreted as a means by which the cult of Becket was promoted (Geijger 1956). Furthermore, this is true of the association with St Lawrence, whose cult was also long-enduring (Gameson 2002). Lastly, the combination of the English saint with St Nicholas, already visible in the painted programme of the lower Church of Anagni (Quattrocchi 2017), is probably an attempt to draw comparisons between the battle fought by Thomas Becket against the reforms brought about by Henry II and the Bishop of Myra's own fight against Arianism[14]. This association could also be an attempt to display the two 'models' of episcopal sanctity—the confessor Nicholas as the archetypal pastoral bishop, and the martyr Thomas as the defender of the Church liberties (Vauchez 1981, pp. 197–98; Gazeau et al. 2015). This makes even more sense given the Italian context, where the mitre was conserved, as the cult of St Nicolas penetrated the West from the south of the peninsula since the end of the 11th century. His relics arrived in Bari in 1087. This serial production of mitres embroidered in English ateliers demonstrates a purposeful adaptation to an Italian demand, unless the mitre was an Italian imitation of the *Opus Anglicanum* (Blöcher 2012, pp. 91–92 and 183; Privat-Savigny 2014). In her works on the Anagni paintings, Claudia Quattrocchi clearly shows how the English saint became a propaganda figure, serving the interests of the Roman Curia and, at the behest of Alexander III, was named as a defender of spiritual and especially papal authority in the face of intrusions from temporal powers (Quattrocchi 2017, 2021).

Thus, in the early stage of the development of the cult of Thomas Becket, the saint's image was explicitly associated with significant Christian figures, seen on each of the mitres. This integrates the saint within Christian martyrial history and within a history of a militant Church. It inserts him into a martyrial and episcopal lineage, into which those bishops who wore these mitres would also be symbolically included. The image embroidered onto the headpiece, in itself a symbol of episcopal authority, visually and efficiently established this spiritual connection, thus becoming a historical *mise en abyme*. According to Caroline Vogt, the prelates responsible for these commissions wanted to draw links between their pontificate and Christian martyrs, past and present. Drawing on a concept established by Michael Greenblat, Vogt interprets the mitres as a 'medium of episcopal self-fashioning'

(Vogt 2010, p. 117). Likewise, in relation to the Jacques de Mitry mitre, Anne Duggan has shown how the context of the Crusades and the existence of a budding cult to Thomas Becket in Acre led the preacher from Liège, named bishop of Acre in 1216, to use his headpiece to show the affliction of Thomas Becket and by doing so invoke the protection of the English martyr—the bishop finding himself on the cusp of the Fifth Crusade, with a perilous and delicate episcopal duty on his shoulders (Duggan 2020, pp. 5–7). In the same way, the acquisition of the mitre offered by the Duchess Ludmilla to the Monastery of Seligenthal most likely proves the very personal devotion she had towards the English saint. The Duchess came from Hungary where Becket's cult was quickly established (Kosztolnyik 1980; Duggan 2020, p. 16). Lastly, we should note that the majority of these mitres were worn or conserved over a long period, a fact that in itself demonstrates the ritualistic and symbolic value of these woven objects[15].

### 3. The Embroidered Panels of the Abegg-Stiftung

The embroidered panels of the Abegg-Stiftung foundation in Riggisberg (Switzerland) are the only surviving examples in the textile world of a narrative cycle dedicated to Becket (Figures 9 and 10). The probable destination of the textile also makes it a remarkable object. The two panels (21 × 49 cm), although acquired on the French market in 1991, are of English origin and embroidered in *Opus Anglicanum*. They date back to the second half of the 13th century and each show three scenes in the life of Thomas Becket, framed under three arches. Hans Christoph Ackerman, who studied the panels in the 1990s, described the exceptional quality of the object's production, composed, among other things, of gold threads which have sadly oxidised (Ackermann 1994a, 1994b). He links the craftsmanship of the panels to the artistic circle surrounding the London court of Henry III at the time of the centenary of the death of Thomas Becket (1270). Ackerman's assessment is that the panels could be pieces of an embroidered sheet that would have covered a rectangular reliquary casket, of which two scenes, located on the short sides, have disappeared, including the murder scene (for a reconstruction scheme, see Ackermann 1994a, p. 280). Much more convincingly, Nigel Morgan suggests that the panels should be identified with the two superimposed registers of an altar frontal comparable to other surviving examples and to Norwegian painted altar frontals (Morgan 2016, p. 37).

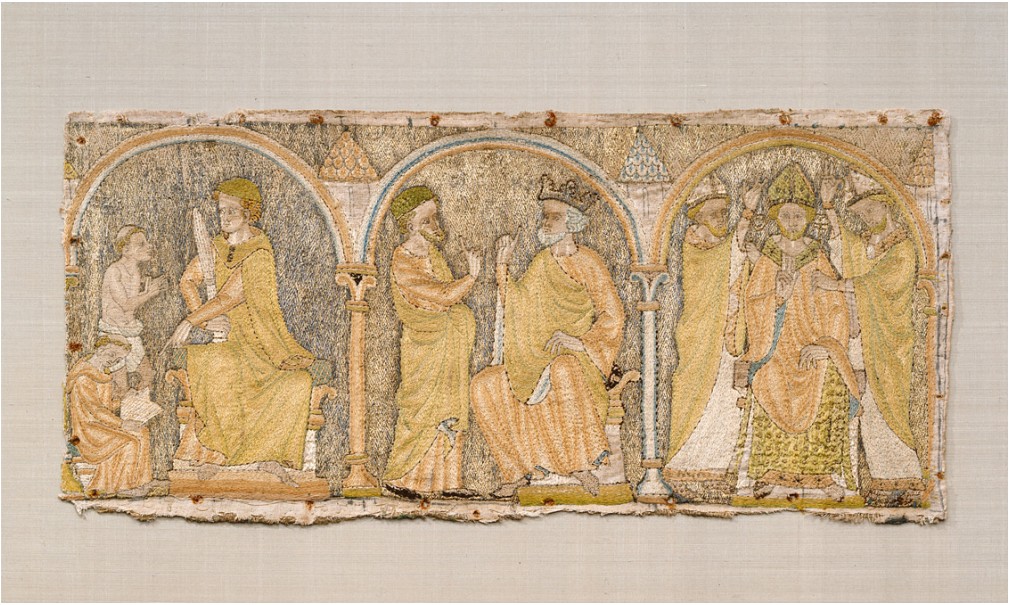

**Figure 9.** First embroidered panel, Thomas Becket's hagiographic cycle, second half of the 13th Century. Abegg-Stiftung Foundation, Switzerland. © Abegg-Stiftung, CH-3132 Riggisberg, 2008. Photo: Christoph von Viràg.

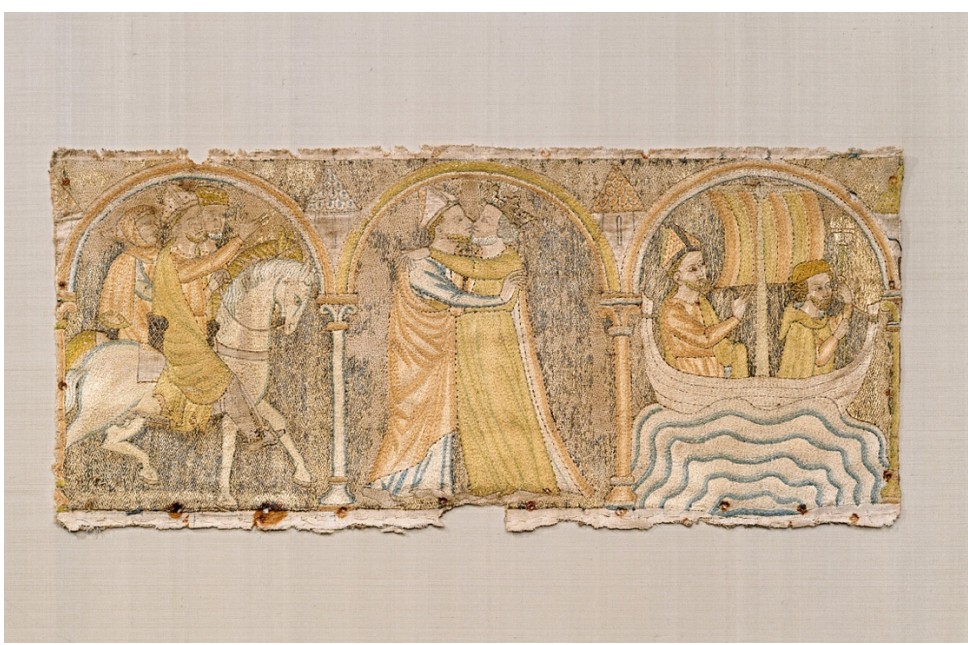

**Figure 10.** Second embroidered panel, Thomas Becket's hagiographic cycle, second half of the thirteenth Century. Abegg-Stiftung Foundation, Switzerland. © Abegg-Stiftung, CH-3132 Riggisberg, 2008. Photo: Christoph von Viràg.

The hagiographic cycle begins by evoking Thomas Becket's formation and would have closed with his death as a martyr. The first panel shows three scenes which begin the account of his life (Figure 9). Under the first arch, Thomas is beardless. He is portrayed in secular dress and is sat on an elaborate seat. He appears to be holding a ferula in his right hand; meanwhile, his left forefinger instructs a tonsured monk to write. Above, a third figure, also tonsured, has a bare chest and legs and wears only a simple loincloth, with his hands clasped in supplication. This image shows Thomas Becket during his period as a cleric or Archdeacon of Canterbury. The following scene shows, on the left-hand side, Thomas in discussion with a king, who can be identified as Henry II. Wearing a crown, the king is seated on a throne, and his right forefinger is raised. On the king's right, Thomas also points the forefinger of his right hand. Becket is now shown with a brown beard, a sign of maturity, in contrast to the white beard and hair of King Henry II which, taking into account the 12 year age difference between the two (Becket being 12 years King Henry II's senior), is a symbol of the king's authority. The two men, as shown in this image, are not yet adversaries and are shown in conversation, although the iconography underscores royal seniority. The third scene shows Becket's investiture as Archbishop of Canterbury. Shown face on, sitting on the episcopal throne, Thomas is mitred and is dressed in sacerdotal vestments. He is framed by two bishops who also wear mitres and who are blessing Thomas. The most likely scene to have followed and which will have been lost is Thomas's meeting with Pope Alexander III at the Council of Tours.

The next three scenes follow on a second embroidered panel (Figure 10). The first scene on the left of the panel shows three horsemen riding in the direction of the action in the embroidery: in the foreground, we see Thomas Becket, wearing a mitre, accompanied by two men. The scene has been identified as the escape from Northampton, following Becket's refusal to sign the Constitutions of Clarendon, an episode that marks the beginning of his exile to France. The second scene shows two men in the act of embracing each other. We can quickly identify Becket, on the left, from his vestment and the episcopal mitre he wears. The second man, on the right, is a king. Should we see this as a scene of reconciliation—a kiss of peace—evoking the meeting organised on the 22 July 1170 by the King of France, Louis VII, under pressure from Pope Alexander III, which took place at the Chateau de Fréteval (hypothesis proposed by H.-C. Ackerman), and which brought

together the English archbishop and King Henry II? It could also be the encounter in Montmartre, on the 18 November 1169. Nevertheless, in both cases, the meeting ended with the refusal of the kiss by Henry II (Petkov 2003, pp. 64–65). The two scenes which highlight temporal and spiritual power have been placed symmetrically in the same place on both sides of the embroidered panels. However, depicting a scene that is not historically attested, this ritualised kiss of peace, which seems now unique in Becket's iconography, is probably a hagiographic choice to make the murder of the English archbishop even more shocking. The third scene on this panel was a direct consequence of the meeting between Henry II and Thomas Becket in France, which would result in Becket deciding to return to England. It shows the saint appearing on a small boat with a billowing sail, which carries him from Wissant to Sandwich. Thomas is accompanied by a second man, probably Herbert of Bosham. Lastly, the final scene of the cycle may have shown the story's dramatic conclusion, with Thomas Becket's assassination in Canterbury.

Although they were produced during the third quarter of the 13th century, the stylistic characteristics of these two panels borrow from the Romanesque art, notably in the detail of the arches under which the cycle evolves. These are also reminiscent of the late 12th century miniatures included in the Harley manuscript (London, BL, Harley 5102, f. 17 and 32). Are there signs of earlier models in this example? Nothing currently allows us to go any further with this hypothesis, although it is highly likely that other narrative examples like this dedicated to the English saint would have been produced during the intense period of diffusion of his cult. In the absence of other examples in the textile domain, the Riggisberg panels can be compared with the narrative cycles conserved in manuscript illuminations (e.g., the illustrated prose life of the saint, the Becket Leaves, from the J. Paul Getty Collection, Wormsley Library, ca. 1230–40, MS 6). Even more pertinently, they can be compared to the monumental windows dedicated to the saint's life at Canterbury, Sens, Chartres, Angers and Coutances cathedrals, all of which were produced during the first half of the 13th century and, therefore, some years before the example in our study (Brisac 1975; Harrison Caviness 1977; Jordan 2009; Jordan 2016). Although the embroidered panels seem summary and reduced by comparison, some of the same scenes from the life of Becket could be seen in these windows (the reconciliation, Becket's return to England). However, we should note that, unlike the earlier cycles which focused on the period from the saint's exile to France to his return to Canterbury and assassination, as well as featuring this period, the embroidered panels also cover the beginning of Becket's career. Although it remains incomplete, the embroidered cycle encompasses a large part of the saint's life, at the very least from when he took up his clerical duties.

## 4. Embroidered Vestments

### 4.1. Maniples

Two maniples are among the oldest preserved textiles of the corpus[16]. One of the maniples conserved in Namur is embroidered in gold thread and dates from the first quarter of the 13th century (Figure 11)[17]. The vestment carries eight saints, four on each face, seen head to toe, who can be identified by descriptions in Latin uncial scripts. The saints appear under arches, atop which sit three smaller, narrower arcs ending in a cross. On the right face, we see the three nimbed apostles Bartholomew, John, and Paul holding his sword. The last figure on this side is St Denis, mitred and in episcopal dress; he blesses with his right hand and holds a cross in his left hand. Three other apostles appear on the left face of the garment: Andrew, James, and Peter and then Thomas Becket who, just like St Denis, appears face on, using his hand to offer a blessing. They wear identical sacerdotal vestments. The iconographic organisation is partly based around pairings, with a classic comparison being drawn between the figures of Peter and Paul, and another, less frequent comparison, at least within the corpus studied, between Thomas and Denis. Here, once again, Becket is associated with another martyr from Christian history, the alleged first bishop of Paris, who was decapitated, and who was a celebrated figure of evangelisation in the earliest phases of the Church. Denis, like Becket but long before, also suffered a form of

torture linked to the head. Furthermore, according to the Becket's hagiographers, sensing his impending death, he commended himself to St Denis (as well as in St Aelfeah) at the moment of his death (Staunton 2006, p. 195; Barlow 1986, p. 248).

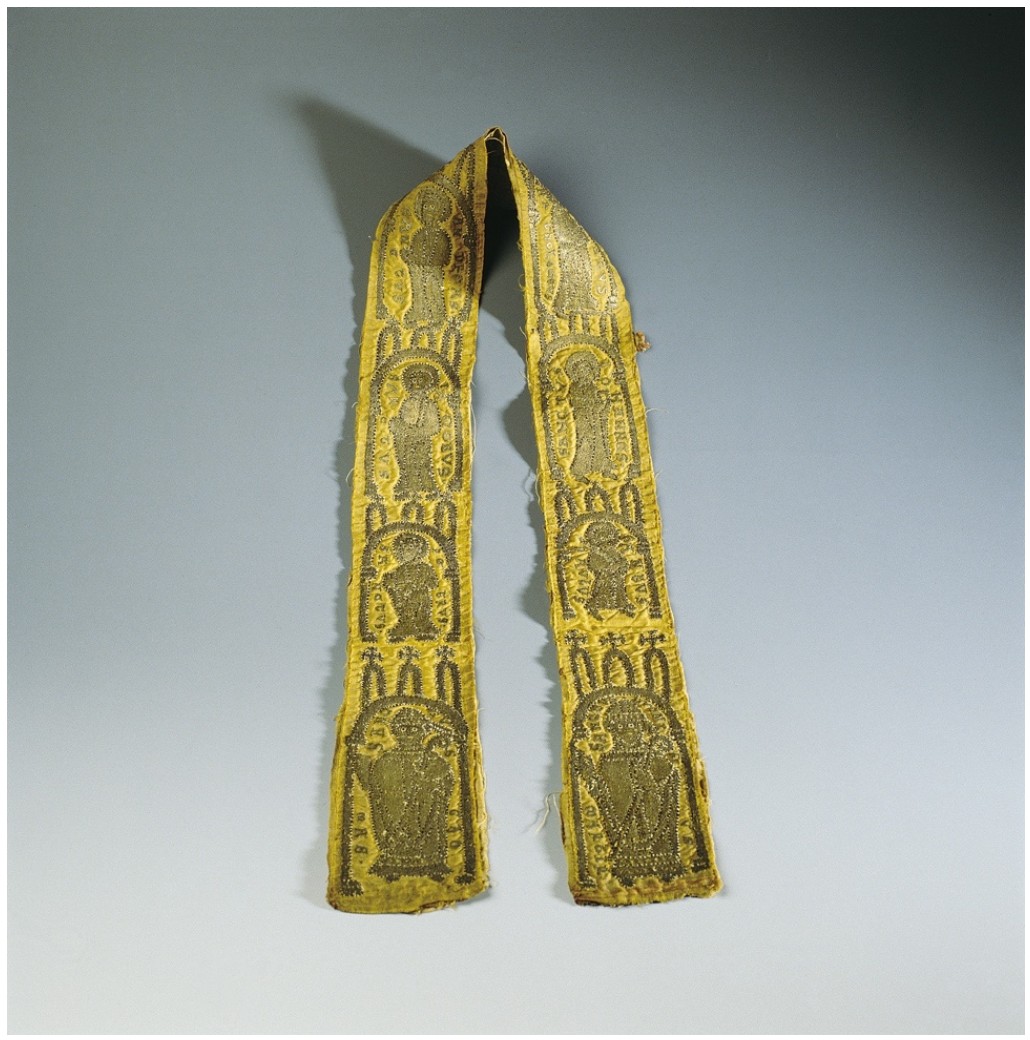

**Figure 11.** Maniple, first quarter of the 13th century. Donated by the Sisters of Notre-Dame de Namur, coll. King Baudouin Foundation, on deposit at the Société Archéologique of Namur and exhibited at the TreM.a—Musée des Arts Anciens, Namur, Belgium. © Hughes Dubois.

The second maniple in the corpus is conserved as fragments at Worcester Cathedral (Christie 1938, pp. 52–54; Coatsworth 2005, p. 17; Owen Crocker 2019, p. 75). Occasionally attributed to Bishop William of Blois (1218–1236), these fabric vestiges have recently been dated to the first third of the 13th century (Blöcher 2012, p. 91, footnote 222). The fragments show the four major prophets, and the trapezoidal edges of the garment would have shown four other figures under arches. Despite the poor condition of the remaining inscriptions, it is possible to establish that Thomas Becket would have appeared among the figures, again full-length, accompanied by St Nicholas, much as on the Anagni mitre.

In his study of the paintings in the Church of Santa María in Terrassa (Catalonia), Carles Sánchez showed that a maniple appears around the left forearms of Thomas Becket and the priest Edward Grim, who appear either side of Christ in Glory, in paintings on the semi-dome of the chapel (Sánchez Márquez 2021, p. 34; Sánchez Márquez and Jiménez 2021)[18]. Sánchez analysed the textile garment as a symbol recalling the pain and sacrifice offered to God, which will be rewarded in the celestial life. Thus, the insignia can both be used to carry the saint's image or be pictured being worn by him within the context of a monumental production.

*4.2. Fragments of Orphrey*

Since 1988, the Victoria and Albert Museum in London has housed two fragments of orphrey that each recount an episode in the life of Thomas Becket (Figures 12 and 13)[19]. Embroidered using the *Opus Anglicanum* technique and dating to between 1380 and 1410, they form the beginning of a narrative cycle. Indeed, the whole orphrey would have most likely included other significant episodes from the English saint's hagiography. Chronologically, the first scene on the orphrey has been identified as featuring the meeting between Pope Alexander III and Thomas who, in 1164, was in exile in Sens. However, it could just as well represent the moment when, during the Council of Tours in 1163, the Pope returned the pallium to Becket, ornament that he is clearly wearing here over his chasuble (Figure 12). Set within an architectural and ornamental framework, at the upper centre of the orphrey, the archbishop, holding his hands in a gesture of submission and prayer, is shown mitred and kneeling before Alexander III who, wearing the papal tiara, occupies the right half of the image. To the left and further back, another ecclesiastic, likely a deacon, holds a ferula in his left hand and a book in his right.

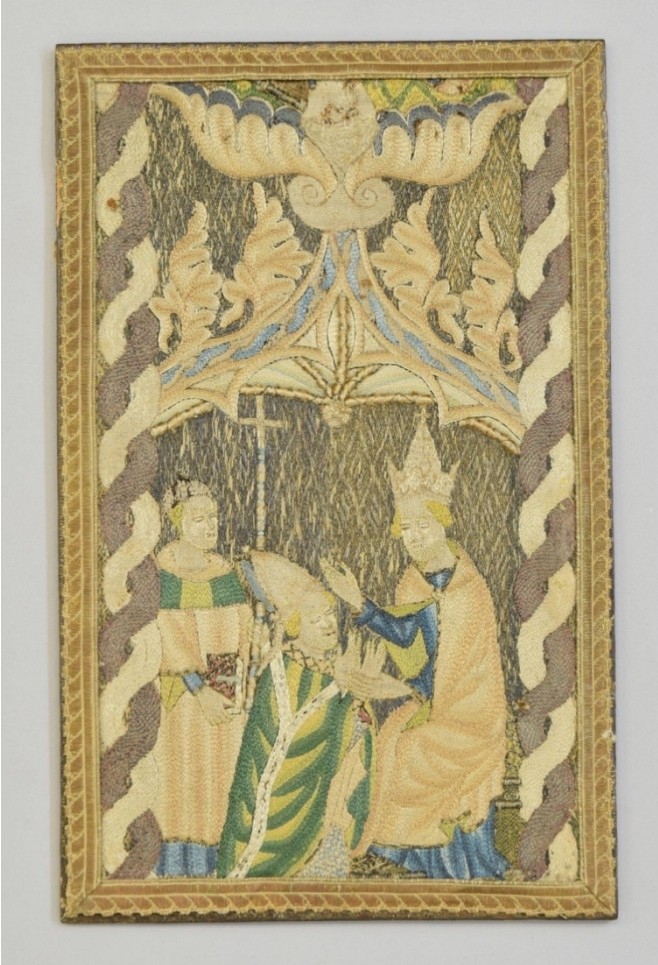

**Figure 12.** Fragment of orphrey, meeting between the Pope Alexander III and Thomas, ca. 1380–1410. London, Victoria and Albert Museum, England. © Victoria and Albert Museum, London.

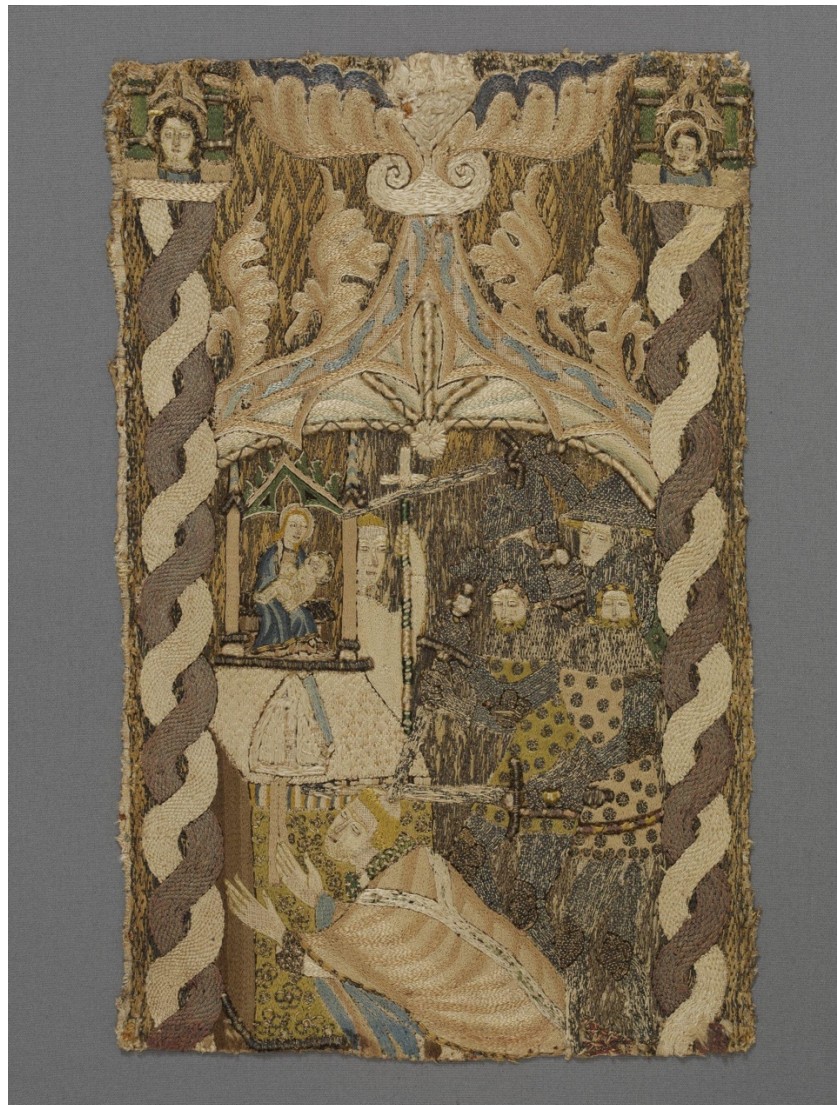

**Figure 13.** Fragment of orphrey, Becket's assassination, ca. 1380–1410. London, Victoria and Albert Museum, England. © Victoria and Albert Museum, London.

The second scene shows Becket's assassination in Canterbury on 29 December 1170 (Figure 13). In an identical frame as the first scene, the saint falls to the ground under the blows of four assailants, armed with swords and shields and dressed in chainmail. Shown here as a discreet witness to the scene, Edward Grim appears in the background, holding a cross—similar in appearance to the previous scene. Dressed identically as in the previous fragment of orphrey, Thomas Becket falls with his hands held out towards the altar, where his mitre is visible. Above the altar, we see an image of the Virgin and Child beneath a baldaquin. This detail is rare in the Becket iconographic repertoire; indeed, it is the only textile example of encountered thus far. In her study, Linda Wooley proposed a link between this Marian image and a passage in the *Vita Sancti Thomae* by William Fitzstephen, in which the biographer reports that Becket spoke the following words before dying (Wooley 1985, p. 267): 'I submit to death in the name of the Lord, and I commend my soul and the cause of the Church of God and St Mary and the patron saints of this church'. (*Vita* I, 154). Becket would, therefore, seem to die in an attitude of total veneration of the Virgin.

Both of the above scenes highlight Becket's episcopal status through his clothing and pay particular attention to his relationship with the Pope. Although we do not currently know where these two fragments of orphrey come from, they probably decorated a chasuble

or cope and would have been worn in an episcopal context[20]. Within the corpus of embroideries, these pieces feature two images that have not been found elsewhere, thus demonstrating the iconographical diversity of the textile corpus.

### 4.3. Copes

The remainder of the corpus, and the largest number of similar items, comprises seven copes. Again created using the *Opus Anglicanum* technique, they were produced in English ateliers where orders were placed either by their eventual owner or as diplomatic gifts, about which prelates were fanatical. The copes are today conserved across the whole of Europe, and they must have played a large part in spreading the cult of Thomas Becket across medieval Western civilisation. Two iconographic types, the most classical and frequent in the Becketian visual repertoire, were used to incorporate an image of the English archbishop into the medieval priest's or bishop's wardrobe[21]. The first type shows Thomas Becket full-length and wearing sacerdotal dress, as an isolated and synoptic image. The second type, narrative by contrast, shows the assassination in Canterbury. The full-length representation of Christian figures, like martyr saints, were common subjects for these semi-circular vestments (Leibacher Ward 2007), which included a hood at the neck of the garment. Very few complete examples have survived.

### 4.3.1. The Anagni Cope

In addition to the mitre that we discussed above, the treasury of Anagni Cathedral includes a second embroidered textile piece that represents Thomas Becket. The Anagni cope, embroidered in *Opus Anglicanum*, dates to the 13th century and appears in the inventory of Pope Boniface VIII (Figure 14)[22]. At an unknown date, the cope was cut up to create two dalmatics, demonstrating the long lifecycle afforded to liturgical vestments and fabrics which could be used and then adapted to serve as different items of clothing. The Anagni cope was reassembled during its most recent restoration in the 1970s, following a reconstruction schema first suggested by A. H. Christie in 1926. In this form, it belongs to the third type of composition used for medieval copes, whereby the surface of the garment is embellished with a geometric grid, using circles, squares, or a quatrefoil pattern, as seen here (Christie 1926, pp. 65–66, pl. II). The motif is repeated at regular intervals across the entire surface of the semi-circular cope. The central scenes of the cope are filled with images of Christ and the Virgin. The remainder of the quatrefoils contain images of martyrs, among whom feature apostles, the very first martyrs, along with medieval saints, such as Becket. Thomas appears on the right half of the cope, given a preferential position not far from the biblical themes (Figure 15). The archbishop is centred, his hands joined in prayer, as he is attacked by three assailants positioned to the left. The attackers, therefore, approach the saint from behind, meaning he falls forward, mitre first, towards a dressed altar. Behind him, Edward Grim is shown holding a ferula. All the classic elements of the martyrdom scene are brought together in this one image.

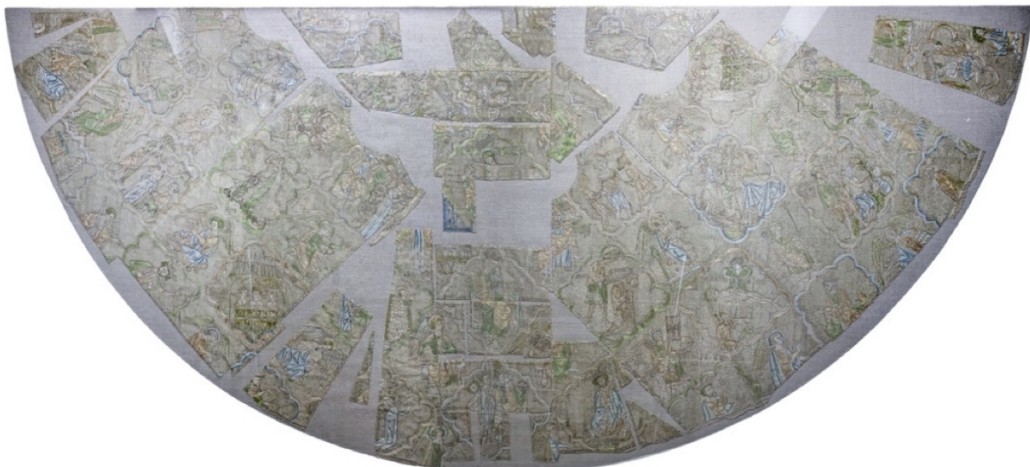

**Figure 14.** Anagni Cope, second half of the 13th century. Ancient treasury of Anagni Cathedral, now in the cathedral museum, Italy. © by concession of the Chapter of the Basilica Cathedral of Anagni. Photo: Graframan.com.

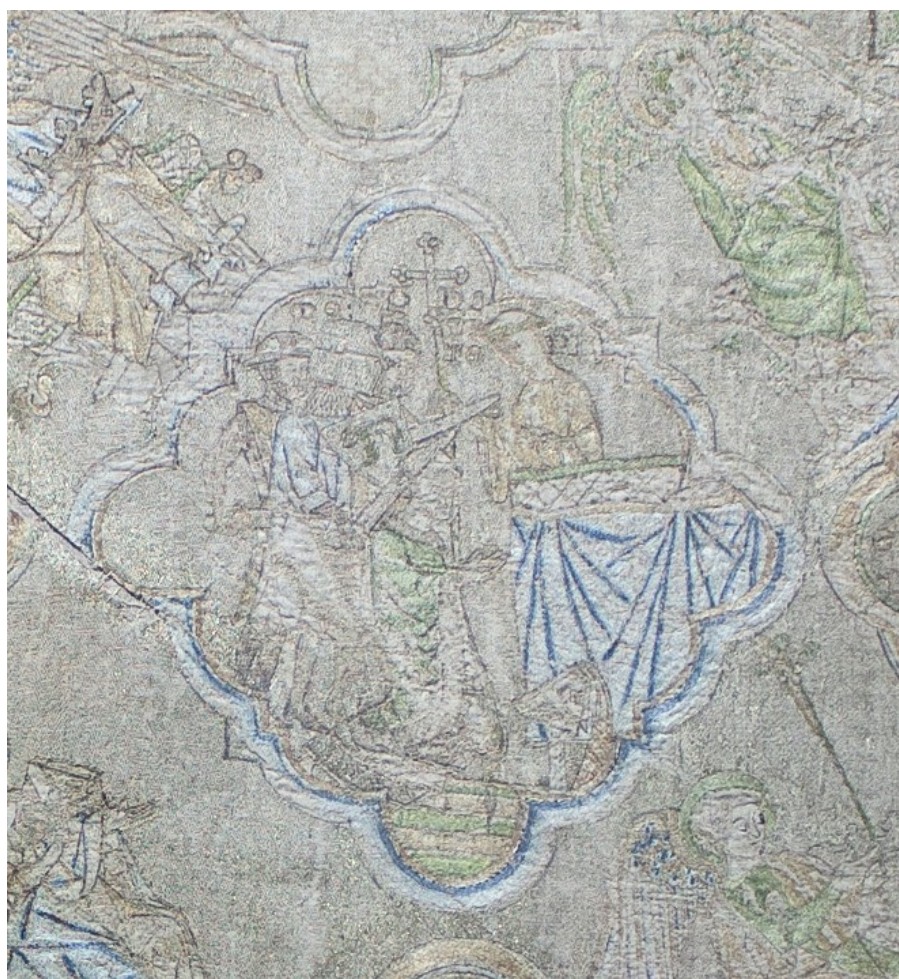

**Figure 15.** Anagni Cope, detail of Becket's martyrdom, second half of the 13th century. Ancient treasury of Anagni Cathedral, now in the cathedral museum, Italy. © by concession of the Chapter of the Basilica Cathedral of Anagni. Photo: Graframan.com.

As with the mitres analysed above, here we see Thomas inserted into Christian martyrial history. The quatrefoil shapes nearest to that which shows Thomas Becket's martyrdom are filled with images of the future patron saint of England, St George, and the

king, Edmund the Martyr. Christie shows how the association of the three saints with links to England (as well as the care given to their representation; they are the only ones on the cope to be provided with their specific attribute) provides clues as to the English origin of the cope (Christie 1926, p. 77).

### 4.3.2. The St John Lateran Cope

Although the scenes of the martyrdom of Thomas Becket depicted on the cope preserved in the treasury of the Lateran do not precisely follow the iconographic schema of the Anagni cope, there are similarities[23]. One similarity is the way in which the vestments worn by the protagonists of the assassination have been represented (Christie 1938, p. 77). As previously, the iconography also integrates images of angels with images of the martyr apostles and saints who are positioned towards the bottom edge of the cope. The vast majority of scenes, appearing under flamboyant braced arches, are dedicated to the Passion of Christ[24] and the last moments in the life of the Virgin Mary; the glorification of the Virgin Mary occupies a central position. Perhaps a gift from King Edward I to the Pope or a commission from John XXII, the Lateran cope was produced a few years after the Anagni cope, probably over the course of the first half of the 14th century (Davies 2016, pp. 77–79). This textile production highlights the persistence of devotion to Becket within the papal curia.

Positioned on the bottom-left edge of the cope, the martyrdom of Thomas Becket shows the archbishop kneeling, face on, arms lifted in a gesture of surprise as his tonsured head is struck by the fatal sword blow from one of Henry II's knight's, positioned to the left of the image (Figure 16). On the right, behind an altar, Edward Grim can be seen holding a ferula in his right hand. Immediately on the left of the scene, we see the Stoning of St Stephen and on the right the martyrdom of St Bartholomew. The ninth century English king, Edmund the Martyr, whose ordeal is also pictured on the Anagni cope, and Thomas Becket are the only medieval saints on the border of the garment, and they are shown alongside the first Christian martyrs. The skilfully organised martyrial iconography of this liturgical vestment demonstrates a willingness to spread the cult of these two English saints across the Western world. Furthermore, it is likely that their images were repeated, this time from head to toe, on the cope's orphrey.

### 4.3.3. The Bologna Cope

Once again conserved in Italy, in Bologna, the third cope (Figure 17) also shows an embroidered image of the murder of Thomas Becket[25]. However, the scene, positioned at the bottom-right edge of the garment, is the only martyrial scene in the iconographic programme. Indeed, the entirety of the embroidered cope is dedicated to the life of Christ, from his Infancy to his Passion, Resurrection, and Descent into Limbo. Dating back to late 13th to early 14th century, the Bologna cope is thought to have been offered by Pope Benedict XI, during his brief pontificate (1303–1304), to monks of the Basilica of San Domenico. Massimo Medica, curator of the Ancient Arts Museums of Bologna, hypothesises that the king of England, Edward I, probably also donated this cope to the pope (Medica 2022).

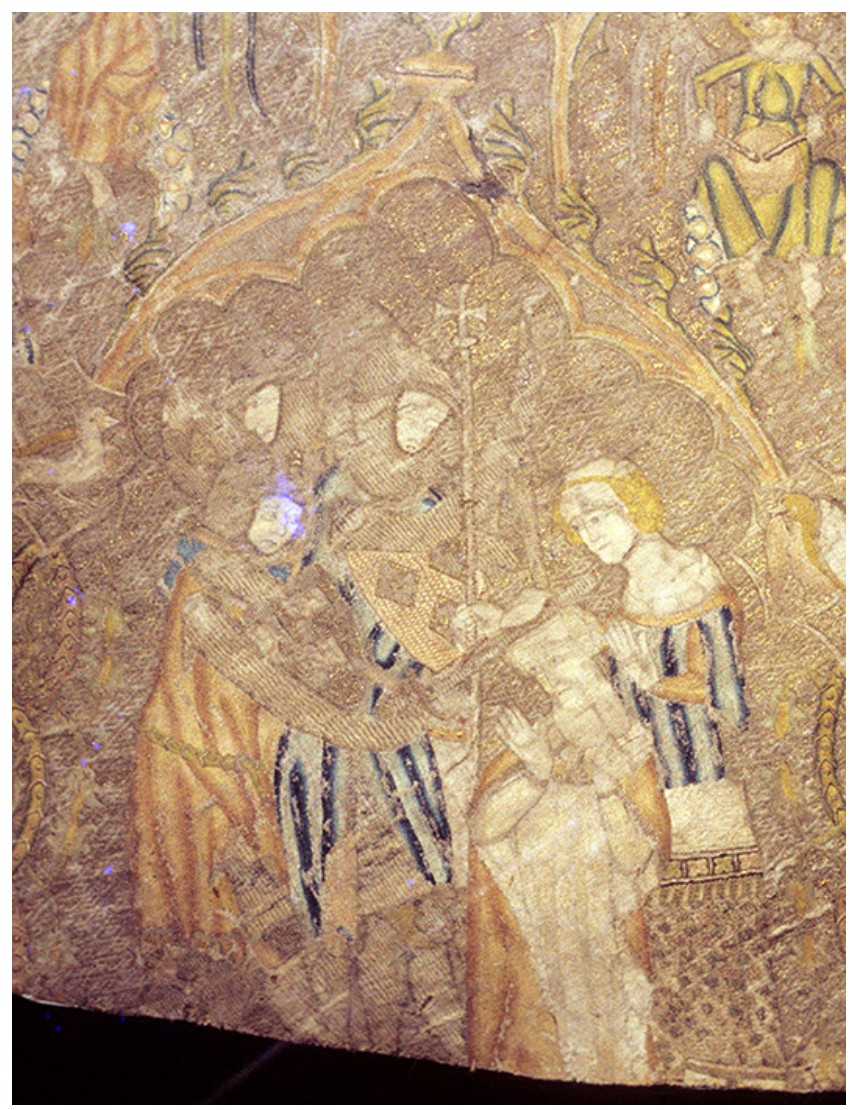

**Figure 16.** St John Lateran cope, first half of the 14th century, detail: Thomas Becket's martyrdom. Roma, Lateran Museum, Italy. © Index of Medieval Art, Princeton.

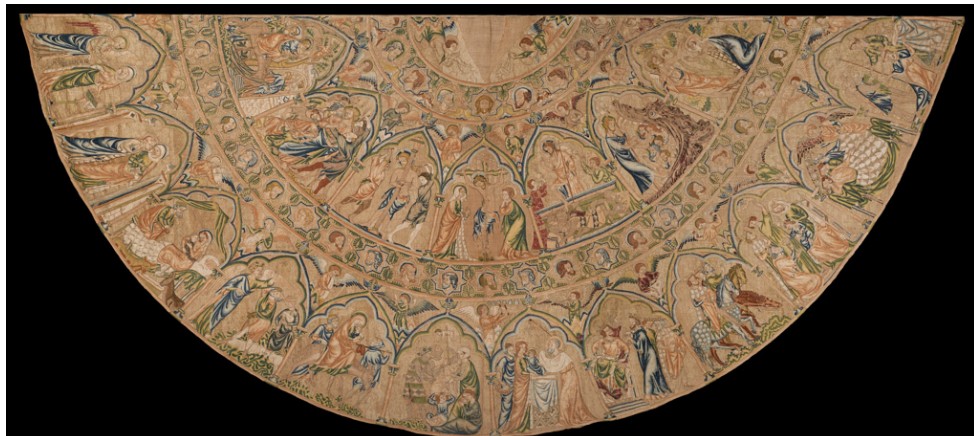

**Figure 17.** Bologna cope, late 13th to early 14th century. Bologna, Musei Civici d'Arte Antica, Italy. © Musei Civici d'Arte Antica di Bologna.

Like the rest of the images on the cope, the martyrdom of the Archbishop of Canterbury is pictured under a multifoil arch (Figure 18). Much like on the two copes described above,

in this scene, we see the knights, four this time, on the left-hand side of the image, while Thomas Becket, bearded and tonsured, is located in the centre and is shown falling to the ground. Meanwhile, behind the altar pictured to the right, the priest Edward Grim holds, as before, a ferula in his right hand and is openly confronted by one of the assailants. In this version, we note the absence of Thomas's headwear, and our attention is instead drawn to the addition of a cloth-covered chalice placed on the altar, which has been prepared for the Eucharist, thus evoking the blood of Christ and his sacrifice. Just as in the previous copes, the integration of the murder of Thomas Becket into the embroidered iconographic programme demonstrates how this garment's patron intended to show an image of spiritual power triumphing over temporal power; here, Becket, a figure of the Church, shows the institution as triumphant. The position of this scene within the rest of the garment is especially noteworthy as it is located on the extreme right of the vestment; therefore, the image would have been visible on the front of the cope when it was being worn. We assume that it played an active part in the ritual power of the garment worn during liturgical ceremonies[26].

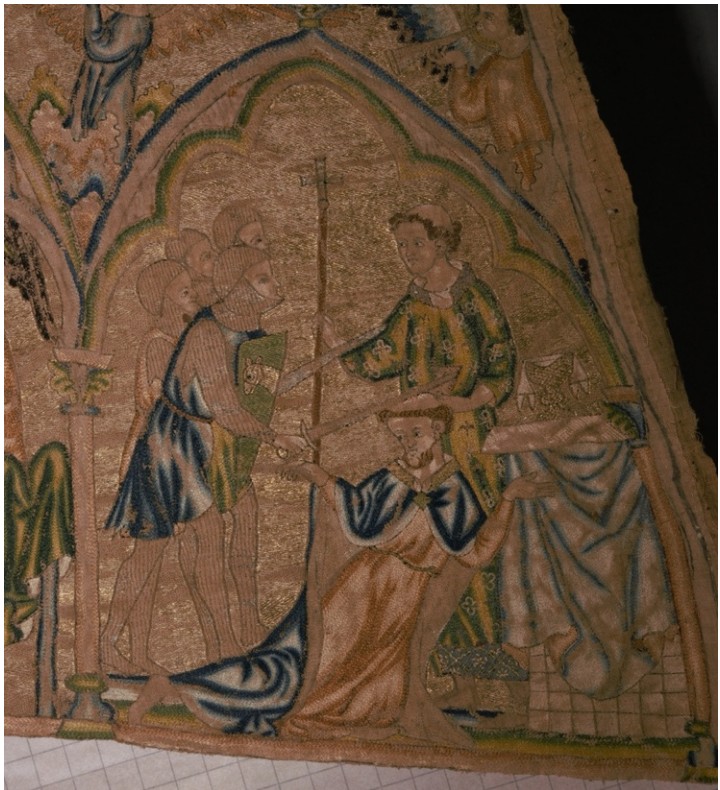

**Figure 18.** Bologna cope, detail: Thomas Becket's murder. Bologna, Musei Civici d'Arte Antica, Italy. © Musei Civici d'Arte Antica di Bologna.

### 4.3.4. The Hildesheim Cope

The Hildesheim cope is the last in this series of copes decorated with images of the martyrdom of Thomas Becket (Figure 19). It was produced around 1310–1320 and originates from Hildesheim Cathedral[27]. Becket's devotion spread quickly on Saxony territory after the saint's death and canonisation, particularly thanks to the patronage of Matilda, duchess of Saxony (1168–1189) and eldest daughter of Henry II, and her husband, Henry the Lion (Bowie 2016, pp. 113–32). The cope is the only surviving example in this corpus of an item that was outside England, as the embroidery is thought to have been produced in a German or Swiss atelier. Similarities with *Opus Anglicanum* can be identified, given that the technique became known throughout the Western world. Although the colours have faded, the 29 medallions show alternating red and green backgrounds, which form the background to a procession of martyrs.

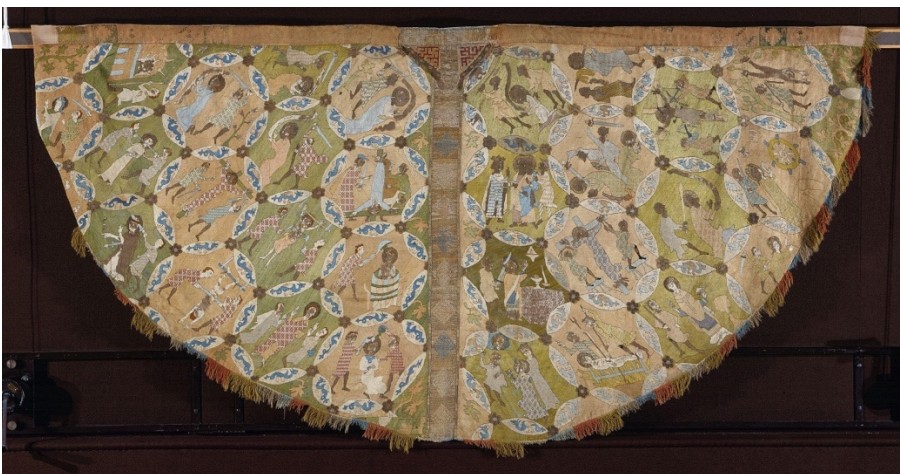

**Figure 19.** Hildesheim Cope, 1310–1320, from Hildesheim Cathedral. London, Victoria and Albert Museum, England. © Victoria and Albert Museum, London.

One such image pictures a lone figure using a sword to attack a haloed bishop, who is in the act of taking the Eucharistic wafer. The bishop is stood before an altar displaying a cup and a candelabra. The scene has, in turn, been identified as either representing the martyrdom of St Matthew or of Thomas Becket[28]. Although the representation of the martyrdom of Becket usually involves the presence of three or four attackers, a small, now lost, reliquary casket from Limoges represented the martyrdom of the saint with only one aggressor[29]. On this basis, the image on the cope could indeed represent the murder of Thomas Becket, without our being able to categorically prove this[30]. We can nevertheless acknowledge with certainty that the scene of martyrdom positioned immediately below this image is the Stoning of St Stephen.

### 4.3.5. The Toledo Cope

The final three copes in the corpus—the Toledo cope, the Butler-Bowdon cope, and the Vic cope—were all produced in England using the *Opus Anglicanum* technique, and all integrate a full-length image of Thomas Becket into their iconographic composition. Unlike the previous examples, they do not include scenes of martyrdom, but instead display individual images of martyred saints. These holy victims accompany an iconographic programme based around the lives of Christ and the Virgin. Dating to 1320–1330, the oldest of the three, from the treasury of Toledo Cathedral (Figure 20), frames the images under trefoil arches and gables[31]. The upper registers are dedicated to themes from the Old Testament and images of apostles, while the third and final row is smaller and is reserved for images of martyrs. Identifiable by Latin inscriptions embroidered onto individual phylacteries, the martyrs are displayed in pairs and are shown triumphing over their persecutors and adversaries, with an overtly victorious attitude. The embroidery of Thomas Becket (Figure 20) shows him in episcopal dress, wearing a mitre and pallium, positioned opposite a crowned King Olaf II of Norway. Becket, who offers a blessing with his right hand and holds a crosier in his left, towers over one of Henry II's knights, who lies at his feet. The man, wearing a coat of armour, is identifiable as Reginald Fitzurse via his shield, which is decorated with a bear. Characteristic of English production, the cope brings together the first Christian martyrs (St Catherine of Alexandria, St Stephen, St Denis, St Margaret, etc.) with a series of English saints, including Edmund the Martyr, Edward the Confessor, Dunstan, Aethelbert, and finally, Thomas Becket.

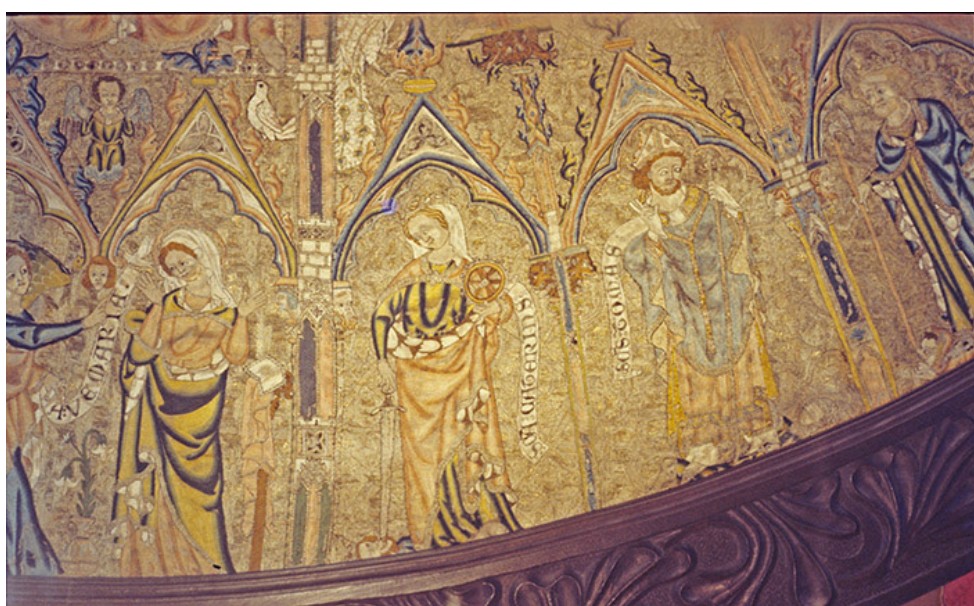

**Figure 20.** Toledo Cope, 1320–1330, detail: full-length representation of Thomas Becket. Toledo, Treasury of the Cathedral, Spain © Index of Medieval Art, Princeton.

The iconography of the cope, featuring saints triumphing over their enemies, has been compared with the 13th century wall paintings in the Painted Chamber in the Palace of Westminster (Browne et al. 2016, pp. 198–200). Now destroyed, the paintings can be studied thanks to their depiction in 19th century copies. These similarities highlight the probability of models, originating at Westminster, circulating among English embroidery ateliers, where they were picked up and integrated into textile art. The Toledo cope has been attributed, over time, to two ecclesiastics with links to the history of Toledo: Cardinal Gil Alvarez Carrillo de Alborñoz, Archbishop of Toledo (1338–1367) and, more recently, Cardinal Pedro Gomez Barroso (died in 1348). A native of Toledo, Cardinal Barroso is thought to have gifted the cope to Toledo Cathedral upon his death (Cros-Guttiérez 2008)[32]. Their familiarity with the curia and the Papal Court of Avignon, where English embroideries were particularly appreciated, undoubtably prompted the acquisition of such a vestment. Cardinal Barroso, a papal legate in France and England, could have acquired the cope during a diplomatic mission to England in 1340 (Browne et al. 2016, p. 201). The prelates' taste for this style of English-origin luxurious embroidered vestment directly contributed to the diffusion of the cult of Becket, in a cathedral where it was already well established since 1177 (Cerda 2016). A chapel dedicated to him was then founded. Moreover, this crucial date marks the beginning of the development of the cult of Becket in the Iberian Peninsula.

### 4.3.6. The Butler-Bowdon Cope

Produced around 1335–1345 and today conserved at the Victoria and Albert Museum, the Butler-Bowdon cope (Figure 21), whose precise origin and patron remain unknow, shows similarities in composition, as well as ornamental and iconographic likenesses, with the Toledo cope, to the extent that we consider them as having been produced by the same atelier[33]. Having been cut into pieces before the 18th century, the cope was later reconstructed; however, some sections from the three concentric rows remain lost; thus, the image of Thomas Becket in episcopal dress, holding a ferula, is incomplete. Scenes portraying the Virgin are displayed along the central band of the cope, and the first two registers are reserved for saints. The last frieze includes images of the apostles. The choice of saints, as well as their positions on the cope, has been skilfully thought out following a system of corresponding pairs organised between the right and left sides of the cope. The pairs are not juxtaposed as they are in the Toledo cope, but are placed on either side of a symmetrical axis which serves as the central band of the vestment. Thus, they appear as

coherent pairings, e.g., the martyrs Stephen and Lawrence, the saints Mary Magdalene and Helen, and the saints John the Evangelist and John the Baptist. As in the Anagni mitre, here too we see a meaningful association between Thomas Becket and the bishop Nicholas of Myra. Lastly, the English origin of the cope is integrated into the garment's composition via the inclusion of two kings: Edward the Confessor and, facing him, Edmund the Martyr.

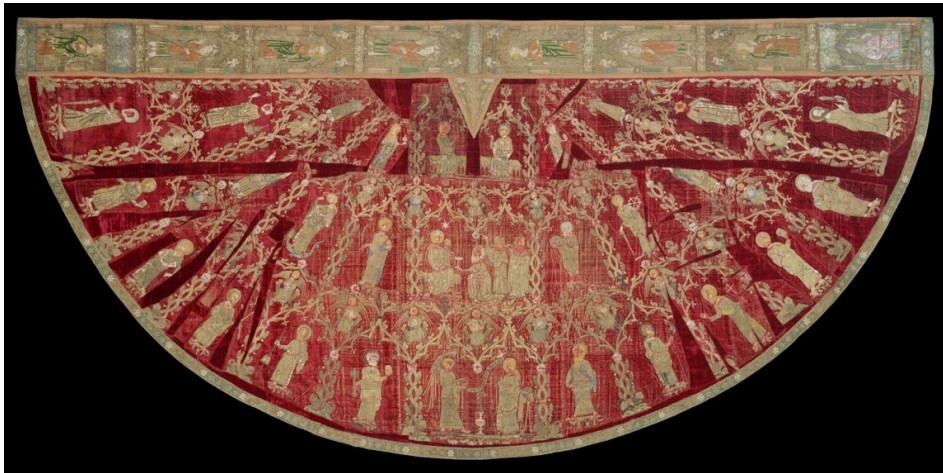

**Figure 21.** Butler-Bowdon Cope, ca. 1335–1345. London, Victoria and Albert Museum, England. © Victoria and Albert Museum, London.

### 4.3.7. The Vic Cope

The final vestment in our study is the renowned Vic cope, originally at Vic Cathedral (Figure 22)[34]. The Becket cult in the territories of the Crown of Aragon, to which Vic city belonged, had spread very quickly after the death and canonisation of the English archbishop. Dating to between 1350 and 1375, the cope has endured a turbulent past, being cut up in the 17th century to be used to make two dalmatics, a book binding and a lectern cover. The pieces were put back together between 1893 and 1899[35]; however, in order to recreate the whole cope, it was necessary to add 28 pieces of red velvet in addition to the 45 original fragments. The current configuration of the cope dates to its restoration in 2008 (Calonder and Wos-Jucker 2008). The cope is an example of an iconographic programme and composition very similar to that of the Butler-Bowdon cope. In addition to the scenes of the Virgin which, as before, occupy the central section of the cope, three concentric registers have been added, showing apostles and saints of the Church (identified by inscriptions). These are, as before, arranged in symmetrical pairs.

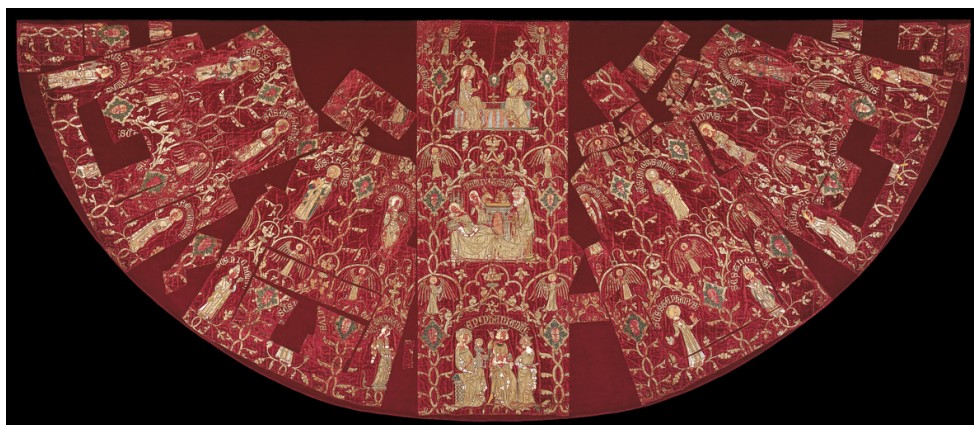

**Figure 22.** Vic Cope, ca. 1350–1375, from Vic Cathedral. Vic, Museu d'Art Medieval, Spain © MEV, Museu d'Art Medieval.

As in the Butler-Bowdon cope, a comprehensive genealogy of saints of the Church can be found on the cope, ranging from the beginning of Christianity up to Thomas Becket, the most recent figure in the iconographic ensemble. Dressed as a bishop, Thomas is mitred and wears a chasuble over an alb. In his left hand, he holds a ferula and he makes a blessing with his right hand (Figure 23). Underscoring its English origins, the cope features royal English saints, including Edward the Confessor and Edmund the Martyr. Once again, the Archbishop of Canterbury is positioned in parallel to St Nicholas of Myra, yet another example of the iconographic pairing that was created in the years just after Thomas Becket's assassination and which would endure for almost two centuries, via the production models of English ateliers. Textual sources make it possible to link this cope with the Bishop of Vic, Ramon de Bellera (1352–1377), who went on to bequeath the cope to the cathedral (Martín Ros 2008, p. 10). Whilst these luxurious copes are, for the most part, linked to cardinal and papal circles (perhaps Avignon in this case), the preciousness of this gift, offered by a lesser member of the high clergy, Ramon de Bellera, to his cathedral can be explained by Catalan cathedral rules, which obliged canons to offer liturgical vestments of some value to their churches (Browne et al. 2016, p. 236).

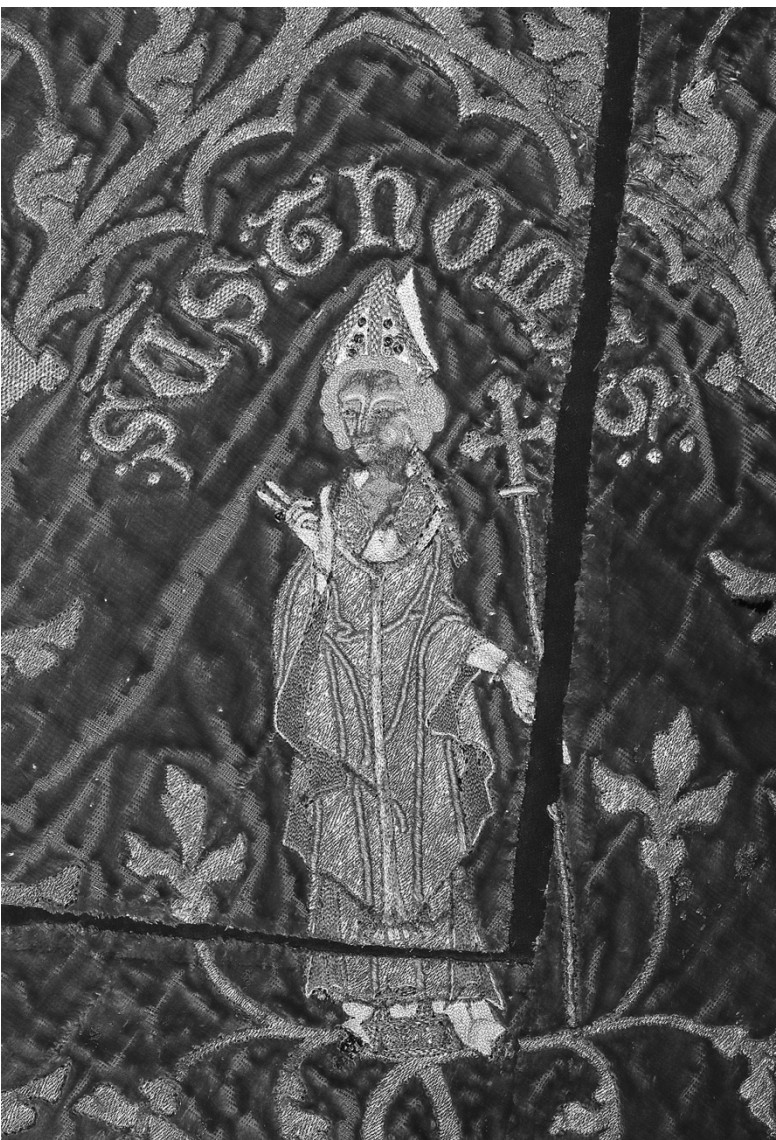

**Figure 23.** Vic Cope, detail of Thomas Becket in full-length, ca. 1350–1375, from Vic Cathedral. Vic, Museu d'Art Medieval, Spain © MEV, Museu d'Art Medieval.

## 5. A Pall Cloth

The final and most recent item in the corpus is a rare survival. It is a pall featuring Thomas Becket that can be attributed to a guild (Figure 24)[36]. The majority of textile pieces that served as coffin coverings for deceased members of the guild have now disappeared, destroyed following the Reformation. Dating to the end of the 15th century or early 16th century and produced using the *Opus Anglicanum* technique, the pall belonged to the Brewers' Hall, an ancient Livery Company of the City of London. The red velvet of the pall forms the background for the Company's symbols (ears of barley, the Company's coat of arms) accompanied by Latin verses from the burial service and the Company's patron saint, Thomas Becket, along with the Virgin Mary. Thomas Becket appears on one of the short sides of the pall cloth, at the opposite end to the Virgin Mary, and is shown seated on a type of throne. Mitred and adorned with a halo, he wears a pallium over an episcopal chasuble. He is shown, as is often the case, blessing with his right hand and holding a ferula in his left. A number of funeral objects such as this, which included an image of the English saint, were made and have unfortunately now been lost.

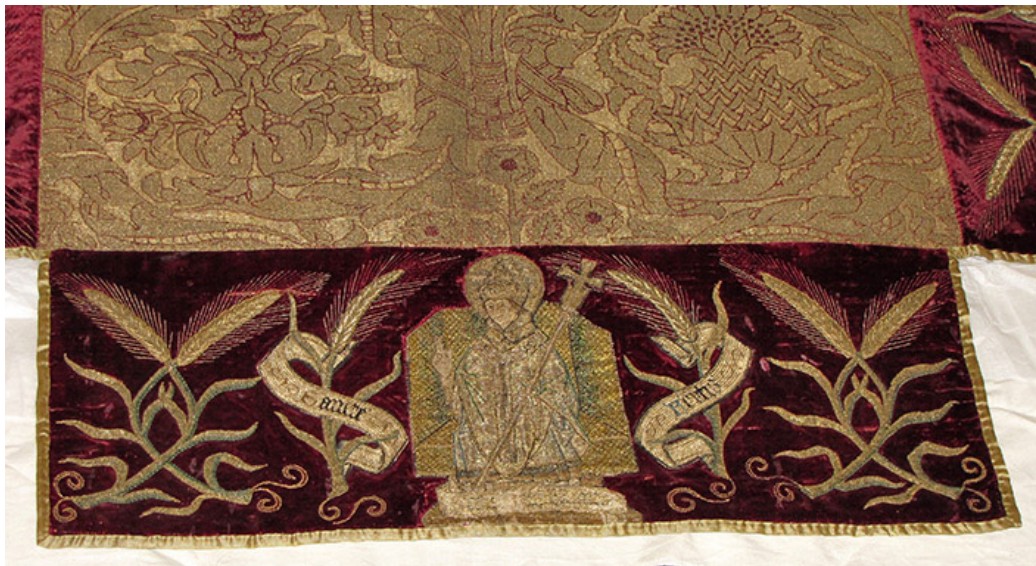

**Figure 24.** Pall Cloth, detail: Thomas Becket, end of the 15th to early 16th century, London, Brewers' Hall, Worshipful Company of Brewers, England. © Index of Medieval Art, Princeton.

## 6. Conclusions

Despite significant losses since the medieval period[37], the material that makes up the textile corpus allows us to understand how the image of Thomas Becket was diffused, via material ranging from liturgical vestments to sacred textiles. All but two items in the corpus belong to the clerical, specifically to the high clerical, wardrobe. The exceptions to this are the pall, which demonstrates the scale of the devotion to the English saint amongst all social groups, and the altar frontal, to which we will return to. For the most part, the items are of English origin, and the examples in the corpus demonstrate an insular, occasionally quasi-serial production. Such is the case with the mitres, which are the oldest examples in our study. The textiles were intended to promote the image of Thomas Becket and his cult, via its diffusion throughout the medieval Western world. However, this type of production quickly reached a public made up of prelates who were interested in the opportunity of wearing a headpiece or vestments (copes and chasubles) that would serve as reminders of the Archbishop of Canterbury. This was the perfect opportunity for a papal curia that, since Alexander III, had tasked itself with promoting Thomas Becket's legacy in the interest of propaganda (Elster 2014; Gardner 2016). Moreover, the presence of the papacy in Avignon during the 14th century participated in the success of *broderie anglaise*, by relaying the technique. By the same token, this presence contributed to the spread of

the cult of the English saint in the highest echelons of the clergy (Paravicini Bagliani 1995, pp. 119–36; Bavoux 2012, p. 160; Descatoire 2019). By offering these precious liturgical textiles to clerical institutions, as was the case for the Anagni, Bologna, and Lateran copes (Elster 2014), the policy of making donations employed by these popes further bolstered the propagation of the cult of the holy martyr.

The role of certain lay dignitaries in this diffusion, particularly of the Plantagenet milieu and their successors at the head of the English kingdom, had also been essential. The Bologna and Lateran copes are, thus, very probably diplomatic gifts from King Edward I (1272–1307).

Among the vestments and accessories of the corpus, some were used as part of the liturgy of the Eucharist (chasuble and maniple) and others, such as the copes, could be used in processions during the divine office[38]. The mitres could be worn in both instances. In this way, wearing the image of Thomas Becket, especially in assuming a narrative representation of his martyrdom, was an effective tool for the ecclesiastic who chose to wear the item to affirm the independence of his power. Although the composition of the copes produced between the end of the 13th century and the first half of the 14th century shows many classical features of imagery[39], the textile pieces that include the martyrdom of Becket could also have been used to underscore the authority gained in the confrontation between the archbishop and the English crown—the battle for the freedom of the Church against a temporal power[40]. This is especially relevant in the case of the mitres, whose visual message was eminently political, as described by Maureen C. Miller: '[w]earing a Becket mitre in the presence of temporal lords might be perceived as something more than a conversation starter' (Miller 2014, pp. 202–6, here p. 205). The same is true for the fragments of orphrey at the Victoria and Albert Museum, and even more so for the altar frontal from the Abegg-Stiftung Foundation. Even though we unfortunately do not know the circumstances of this commission, its symbolic and political impact and significance can be compared with contemporary monumental apse decorations.

We have shown how Thomas Becket appeared on embroidered vestments and textile pieces, ranging from commissioned items to purchases and diplomatic gifts, which belonged to bishops, cardinals, and popes (Gardner 2016). The late 12th century saint was almost one of their counterparts, and his image served their purpose as servants of the Church. From the end of the 12th century until the 14th century, Becket was elevated to the position of defender of the Church liberties against the monarchical institution. He became the archetype of sanctity, particularly the universal model of the martyred bishop, of the bishop who opposed the monarchical State, until the end of the Middle Ages (Vauchez 1981, pp. 197–203). It is, therefore, not surprising to see him represented here on episcopal or pontifical vestments. The representation of the martyrdom of the English saint which shows him, in most cases, in front of an altar prepared for the celebration of the Eucharist (despite accounts of real events) created a parallel between his sacrifice, linked to his episcopal duty, and the sacrifice made by Christ[41]. Becket's image probably serves here to remind the bishops of the sacrifice that the saint's martyrdom entailed, as a sacrifice parallel to that which medieval bishops made of their lives and the risks associated with their ministry. Becket was a new model of a saint, coming from the ranks of the clergy, and no longer from aristocratic lay circles.

In consequence, wearing the image of Thomas Becket contributed, via its iconography, to the self-glorification of the prelate, i.e., the bishop (Joubert 2006, p. 10) who possessed the object, and, by extension, to the construction of an episcopal *memoria*[42]. Furthermore, it could be considered as a means by which to update this episcopal *memoria*, given the chronological proximity which linked Becket, a model of sanctity, with the bishop in possession of the garment. Some ecclesiastical figures, including Jacques de Vitry and the bishops of Anagni, enjoyed personal links with the English saint and worshipped him at times over the course of their lifetime. Unlike other artistic creations displaying images of Thomas Becket (monumental painted programmes, stained glass, reliquaries, etc.), the vestments, thus, offered their patron and/or owner a unique opportunity to possess an

object of special identity, which chimed with their own duty and individuality, serving as a powerful and effective means of expression and corporeal identification.

**Funding:** This research received no external funding.

**Institutional Review Board Statement:** Not applicable.

**Informed Consent Statement:** Not applicable.

**Data Availability Statement:** Not applicable.

**Conflicts of Interest:** The author declares no conflict of interest.

## Notes

1    On the life of the saint and the diffusion of his cult, cf. (Barlow 1986; Duggan 2007, 2012; Webster and Gelin 2016; Slocum 2018; Jenkins 2019; Beer 2021).

2    This article does not consider technical questions on textiles or embroidery, which the author does not specialise in.

3    The *Opus Anglicanum* (Latin for English work) is fine needlework of medieval England mostly for liturgical use on vestments (such as copes, orphreys, mitres, chasubles, etc.), altar frontals, palls, hangings, etc. Famous all across Europe since the 12th century, these expensive embroidery pieces were usually made, using silk and gold or silver threads on linen grounds, or later on luxurious velvet.

4    In some cases, they may have been hidden by families of English Catholics: Wooley (1985, p. 265).

5    There are several studies on the iconography of Thomas Becket. The first being by Tancred Borenius (Borenius 1932). For more recent analyses, cf. (Nilgen 1996; Gameson 2002; Little 2002; Nickson 2020).

6    Conversely, our study does not look at the garments worn by, or which came into contact, with Thomas Becket. For more information on the subject, cf. (Mercier 2003; Coatsworth 2012; Shalem 2017).

7    The surviving corpus is probably larger than detailed in this article. Some full-length representations of the bishop lack an inscription or distinctive symbol, rendering it impossible to accurately identify the figure.

8    Inv. N°TO27, collection Fondation Roi Baudouin (18.8 × 28 cm). (Christie 1938, p. 60, cat. 20, pl. XIX; Blöcher 2012, pp. 283–84, cat 59).

9    Inv. N°T.17. (17.3 × 28 cm), cf. (Christie 1938, pp. 60–61, cat. 21, pl. XIV; Durian-Ress 1986; Blöcher 2012, pp. 282–83, cat. 58).

10    Inv. N°. B 561. (21.5 × 28.5 cm), cf. (Christie 1938, pp. 61–62, cat. n°22, pl. XV; Blöcher 2012, pp. 334–35, cat. 84; Beaulieu and Baylé 1973; Brel-Bordaz 1982). One of the lappets of this mitre now belongs to the collection of the Victoria and Albert Museum in London: Browne et al. (2016, pp. 124–25).

11    Inv. N°12/00202834, scheda N° 22 (22 × 20 cm), cf. (Blöcher 2012, pp. 183–84, cat. 5).

12    For an attempt, cf. (Blöcher 2012, p. 90).

13    St Nicholas is linked to Becket in sources as early as the end of the 12th century. During a storm at sea, which took place during the Third Crusade (1190), Thomas, accompanied by the St Nicholas and by Edmund the Martyr, appeared to the English crusaders three times: Roger of Howden, Gesta regis Henrici secundi (wrongly attributed to Benedict of Peterborough), ed. W. Stubbs, 2 vols (Rolls Series, XLIX, 1867), II, 116; Chronica magistri Rogeri de Houedene, ed. W. Stubbs, 4 vols (Rolls Series, LI, 1868–71), III, 42–43, cited in Duggan, p. 6. On the relationship between the cults of St Nicholas and Thomas Becket, see also (Cavero 2012), and on the martyrdom iconography of medieval mitres, see (Vogt 2007).

14    It should be noted that the stained glass dedicated to Thomas Becket in Chartres Cathedral is associated in the same chapel (Chapel of the Confessors, in the south of the ambulatory) with a stained-glass piece dedicated to St Nicholas, among others. This is also the case in the lower church at the Abbey of Sta Scholastica, in Subiaco, where the paintings link St Stephen with St Nicholas and Thomas Becket (13th century): Cerone (2021). About the visual responses to the sainthood of Thomas Becket in Italy, see (Cipollaro and Decker 2013).

15    The Braga mitre (cathedral museum, Portugal), for example, dating back to the beginning of the 13th century (which brings together the martyrdoms of Stephen and Lawrence) was found in a tomb belonging to a bishop from the middle of the 14th century: (Vogt 2002; Vogt 2010, pp. 119–22; Blöcher 2012, pp. 207–9, cat. 17).

16    The maniple is a 'liturgical symbol for the bishop, priest, and deacon (eighth century), and later for the sub-deacon serving at the altar (from the ninth century). It resembles a strip of material, generally matching the colour of the chasuble. It derives from an item of Roman clothing used to wipe the face, and which was worn less out of utility and more for its ornamental character. When the stole was no longer used as a shroud, the maniple took its place. It, therefore, became an ecclesiastical symbol towards the end of the sixth century, but did not become fully liturgical until the ninth century. Up to the 11th century, it was held in the left hand. Nowadays, it is worn on the left forearm. From the 12th century, the ends of the garment were extended until, in the 16th century, they became spatula-like in shape. The maniple is a symbol of good deeds, vigilance, and also penance'; cf. (Bavoux 2012, p. 842).

[17]   Inv. N°TO29, collection Fondation Roi Baudouin (119 × 8 cm). The patron of the textile piece was probably Jacques de Vitry.

[18]   Milagros Guardia identified the two figures in the upper area of the paintings as Thomas Becket and St Stephen (Guardia 1998, p. 54)

[19]   Inv. T.5.-1988 (33 × 21 cm) and T.5A-1988 (29 × 22.5 cm), cf. (Kurth 1935; Wooley 1985; Owen Crocker et al. 2012, pp. 64–68; Browne et al. 2016, pp. 245–46).

[20]   Like that which is conserved at the Victoria and Albert Museum in London (Inv N°935–1901) and which dates to the first quarter of the 14th century, in which the saint, unlike here, is shown full-length; cf. https://collections.vam.ac.uk/item/O361711/chasuble-unknown/ (accessed on 13 January 2022).

[21]   The cope derives from civilian clothing, dating back to antiquity, and was conceived as protection against bad weather, especially rain; the raincoat, a round and relatively large coat, also has a hood. In the Middle Ages, 'its benediction was not obligatory. This exception is perhaps down to the specific history of this ornament, which was originally used for processions, and therefore outside the church. It was perhaps not strictly a sacerdotal vestment as it was only worn by the officiating priest or bishop during liturgical activities where consecration did not take place'; cf. (Bavoux 2012, p. 129).

[22]   Inv. N°12/00202817, scheda N°5 (140 × 325 cm), cf. (Elster 2018, pp. 321–35).

[23]   158 × 336 cm, cf. (Christie 1938, pp. 149–52, cat. 78; King 1963, pp. 40–41; Linnell 1995, pp. 183–223).

[24]   Only three feature the Childhood of Christ: the Annunciation, the Nativity, and the Adoration of the Magi.

[25]   Inv. N°2040 (149 × 326 cm). The bibliography of this cope is extensive; therefore, we refer to the most recent references: (Bussolati 1993; Owen Crocker et al. 2012, pp. 82–83; Gardner 2016, pp. 213–15; Browne et al. 2016, pp. 176–79). A monograph is currently underway, under the direction of the Musei Civici d'arte Antica di Bologna and Michael A. Michael (2022). See references below.

[26]   On the ritual power of vestments within medieval liturgy, see (Glodt 2019). *Carne amictus. Vêtir le prêtre, parer l'autel au temps de l'hyperréalisme eucharistique (fin du XIIIe-début du XVIe siècle).* (Dressing the priest, adorning the altar in times of Eucharistic hyperrealism) Archivist-palaeographer thesis. Paris, Ecole nationale des Chartres. Thesis statement at http://www.chartes.psl.eu/fr/positions-these/vetir-pretre-parer-autel-au-temps-hyperrealisme-eucharistique-fin-du-xiiie-debut-du (accessed on 20 January 2022).

[27]   Inv. N°17–1873 (146 × 298 cm). (Kroos 1970, pp. 142–43, cat. 81; Brandt 1991, pp. 168–69. cat. 61).

[28]   On the British Museum website (https://collections.vam.ac.uk/item/O113500/the-hildesheim-cope-cope-unknown/; accessed on 20 January 2022), the scene is identified as the martyrdom of Thomas Becket. Kroos sees the martyrdom of St Matthew (Kroos 1970, pp. 66–69). One of the most recent publications on the vestment does adjudicate in the question of identification, although it suggests that the first hypothesis is interesting. This is how the suffering of Becket is linked with the suffering of another medieval saint who appears on the same register: the Dominican Friar of Verona, Peter Martyr: (Coatsworth and Owen Crocker 2018, pp. 110–13).

[29]   Reliquary of the martyrdom of St Thomas of Canterbury (18.7 × 19 cm), originally conserved in Warsaw (Poland), department of National Heritage: (Świeczyński 1988). This is also the case of a reliquary casket attributed to Becket and preserved in St Lawrence Church, at Le Vigean (France).

[30]   Perhaps we should consider the scene as an archetypal image of martyrdom linked to the altar? On this topic, cf. (Gauthier 1975).

[31]   Museo de Tapices y Textiles de la Catedral (169 × 322 cm), cf. (Thomas 2016, pp. 206–8).

[32]   The coat of arms of Cardinal Gil Alvarez Carrillo de Alborñoz would have been embroidered onto the cope, in addition.

[33]   Inv. N° T.36–1955 (162 × 287 cm). We cite only the most recent references: (Owen Crocker et al. 2012, pp. 105–6; Browne et al. 2016, pp. 213–16; Coatsworth and Owen Crocker 2018, pp. 114–19).

[34]   Inv. N° MEV 1430 (132 × 324 cm). On the work, cf. (Martín Ros 2008; Browne et al. 2016, pp. 233–38).

[35]   The reconstruction was carried out on the initiative of Louis de Farcy, from Angers, who was one of the precursors of the study of textiles in the nineteenth century. Published at the end of the century, his work on medieval embroidery remains a source of reference (De Farcy 1890).

[36]   (Christie 1938, n° 268; King 1963, p. 58, n° 151). Reproductions can be viewed in the Princeton University Index of Medieval Art: https://ima.princeton.edu/digital-image-collections/view/44428 (accessed on 20 January 2022). About the relationship between the citizens of London and their patron saint, as well as the strong devotion to Becket, see (Jenkins 2020).

[37]   We should also add several textual references, which refer to vestments or liturgical pieces on which Thomas Becket appears, to our corpus.Borenius (1932, p. 83) mentions a chasuble from the 15th century which is supposed to be held in the collections of Stonyhurst College (but which I was unable to trace).An inventory from 1388 from the 'Vestry of Westminster Abbey' describes an altar frontal which featured the Nativity, the Passion of Thomas Becket and the life of St Edward. It also seems to be mentioned in an inventory from 1540 but the Becketian episode had evidently disappeared at this point (Borenius 1932, p. 88).The inventories of 1245 and 1295 from St Paul's Cathedral in London list a stole and a maniple which featured, among figures of apostles and prophets, St Thomas and Nicholas. (Lehmann-Brockhaus 1955–1960, Bd 2, p. 137).

38     One of the possible development axes of our study would be to examine all these vestments in light of liturgical offices for the cult of the saint who was annually commemorated throughout Europe with specific prayers, frequently with special music, another way to keep his memory alive; on this topic, see (Reames 2000; Emery 2020).

39     Nigel Morgan discusses 'multipurpose imagery' with regard to these copes, whose standardised repertoire was popular among the high clergy with whom this luxurious production found great popularity (Morgan 2016, p. 32).

40     The battle against secular power and the persecutions linked to this combat are a central theme in episcopal hagiography as early as the Gregorian period. We can see that, on some copes, Becket is compared with models of royal saints (e.g., Edward the Confessor and Edmund the Martyr). For more details on the symbolic and political weight of the dedications to Thomas Becket in churches and on altars, as well as representations of the saint in Medieval Italy, cf. (Bottazzi 2011).

41     On the assimilation of the martyrdom of Becket to Christ's sacrifice, and the iconographic shift that it results in, cf. (Aurell 2003).

42     For more information on the creation of a common *memoria*, which allowed for the construction of a collective self-conscience, we can consult the reference studies (Halbwachs 1925, 1950; Oexle 1992, 1995; Lauwers 2002; Destemberg 2015).

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
