# Peer review of "Embroidering the Life of Thomas Becket during the Middle Ages: Cult and Devotion in Liturgical Vestments"

_arts_

Round 1
Reviewer 1 Report
This is a very interesting and well-considered article that will provide interesting advances in two lines of interest: Becket's iconography and the history of liturgical vestments of late medieval bishops. The subject has been approached through the study of a series of paradigmatic examples that constitute the main corpus of works. In addition, the article shows another series of examples in the form of complementary images which it is suggested being included in the main body of the article, if the rules of the journal allow it.
Being a work suitable for publication practically in its current state, it is suggested that the author could enrich his or her conclusions with a little more depth, so that the article is not a mere iconographic compilation of the pieces, classified by typology. To this end, the author is asked some questions that may help him or her to expand on the ideas in his or her conclusion:
- The liturgical vestments chosen were part of the pontifical vestments, i.e. those proper to bishops. Even some of the non-restrictive ones, such as the copes or the maniples, might suggest that they were intended for use by certain bishops. Perhaps the life of the saint is being used as an example of holiness for other bishops? Perhaps Becket's image serves to remind bishops of the sacrifice that the saint's martyrdom entailed, as a sacrifice parallel to that which medieval bishops made with their lives? In this sense, it is worth considering the meaning of some liturgical prayers recited in the Middle Ages and later at the time of vesting before mass.
- It would also be interesting to try to link the places where textiles were produced or the places for which they were produced with a particular devotion to the image of the saint: were there chapels, images dedicated to the saint or references to his existence in the past in these places? It would be very useful in this respect to check the liturgical books of the time in these places to see how, if at all, the feast of this saint was celebrated.
Finally, the critical apparatus of the article is quite complete and for the definition of the corpus of images it may be necessary to contribute some complementary study that evokes other previous works on each work. Dr. Carles Sánchez has recently coordinated a volume of this journal devoted to the figure of Becket and his image. We recommend citing some of these texts, as well as other works by this professor which deal with the figurative construction of the saint in medieval images. On late medieval liturgical vestments and their visual codification, some references by Dr. Pazos-López can be enriching. Other works on Thomas Becket can enrich the specific perspective of the works discussed: Rachel Koopmans, Richard Gameson, John Jenkins, Katherine Emery, among others.
Author Response
Above all, thank you for all the comments and compliments you have made on my article.
In the first paragraph, you suggest to include complementary images in the main body of the article but I am not sure to understand about which series of examples you are talking about ?
I fully agree with what you write in the third paragraph. In the conclusion, I indicate that the choice of including Thomas Becket in the iconography of these vestments for bishops was precisely to serve as a model of sanctity for them and to create a parallel between the life of the English saint and these prelates, between the sacrifice of the latter and the importance of their duty (I suggest that wearing such a vestment could go as far as body identification). This is what I have tried to show in the last two paragraphs of the article but I can try to make it even clearer.
4th § :
Unfortunately, the precise provenance of the majority of these textiles is unknown. On the other hand, when I have been able to link them to a patron and/or a specific place of reception, I have tried to show the relationship between the adoption of the image of Thomas Becket and the cult dedicated to him: this is the case, for example, for the textiles from Anagni, Sens, or those attributed to Jacques de Vitry. The link to the prayers you mention, both in the rituals of vesting before mass, and in the local devotion to the saint, is also extremely interesting, but in the present state of my research, I cannot unfortunately say much more.
5th paragraph : I will include in the article the recent works you mention. With regard to the corpus' examples, I have deliberately chosen to indicate only the most recent references (in which older works are included) to avoid overloading the article's critical apparatus.
With my best regards
Reviewer 2 Report
l. 124 Grim is often absent from the smaller Limoges chasse depictions - where space necessitates a reduction in the number of knights he is often left off as well. (NB A key group of images that Grim is absent from is Flemish miniatures depicting the scene (see Hampson and Jenkins 'A Barber Surgeon's Instrument Case...' Arts 2021 10:3 49). But given the dating of these mitres that may be too fussy a point.)
l. 159 Given the episcopal context of the mitre, could we also be looking at an attempt to display the two 'models' of episcopal sanctity - the confessor Nicholas as the archetypal pastoral bishop, and the martyr Thomas as the defender of church liberties? Just a thought, really (and see Vauchez, Sainthood, pp. 168ff for the 'Becket Model')
ll. 202-203 While Ackerman may have thought that the creation of the panels coincided with a relic translation in 1270, there is absolutely no evidence that any such translation occurred in that year at Canterbury Cathedral, at least (see Jenkins, 'Modelling the Cult of Thomas Becket' JBAA 2020 pp. 108-9). Not much information about the 1270 Jubilee survives, but enough in-house Canterbury material does to be able to state firmly that nothing major occurred. The greatest authority on the Jubilees, Raymonde Foreville, suggests that Henry III did not even go to Canterbury in 1270, and Archbishop Boniface of Savoy was on the Continent, which points to no major translation of relics (Foreville, Le Jubilé de Saint Thomas Becket 1958, pp. 13-14). This suggests that the altar frontal theory is most likely.
ll. 247-51 This is a very interesting scene, as it is explicitly stated that Henry II refused the kiss of peace to Thomas (see Petkov, The Kiss of Peace: Ritual, Self, and Society 2003, pp. 64-5). The scene could either be Freteval on 22 July 1170 or Montmartre, November 18 1169, both of which were meetings between Henry and Thomas, and both ended with the refusal of the kiss. Very interesting either way - and would serve to make Becket's murder all the more shocking, which would be worth mentioning in the article.
ll. 298-301 More relevant to the Becket/Denis link is that according to the hagiographers Becket commended himself to St Denis at the moment of his death (Barlow, Thomas Becket, p. 248)
ll. 315-21 Is Sanchez right? It seems extremely unlikely that an English unsanctified priest would be painted next to a Christ in majesty, regardless of what he'd done. Milagros Guardia much more convincingly suggests it's St Stephen with Becket in that painting, 'Il precoce approdo dell’iconografia di Thomas Becket nella penisola iberica. Il martirio di Becket o il racconto di una morte annunciata.' in I santi venuti dal mare. Atti del Convegno Internazionale di Studio (Bari-Brindisi, 2005), edited by Maria Stella Calò Mariani. Bari: Adda Editore, 2009, pp. 35–56 (at p. 54). I'd suggest omitting this paragraph as it doesn't really add anything and is based on some fairly flimsy interpretation.
ll. 605-6 Is there any idea of how this image of Becket survived, given that it is on an important livery item in the centre of London! Was the name of St Thomas unpicked but the image (of a generic bishop) left?
Author Response
Thank you for all your comments.
l. 124 : You are right, I will integrate the comment
l. 159 : I agree with you and I am going to incorporate this comment.
II. 202-203 : I will amend the paragraph accordingly.
II. 247-251/298-301 : ok.
II. 315-321 : I am quite convinced by Carles Sanchez interpretation. But I can quote also Milagros Guarda's suggestion, in a footnote.
ll. 605-6 : Unfortunately, I have no idea and the documentation I have been able to access does not specify this. I have tried several times (once again last week) to contact the Brewers' Hall where the pall cloth is conserved in London but my attempts have been unsuccessful. Most probably, the pall cloth has been hidden during the modern times.
With my best regards.
Reviewer 3 Report
I think this is a major contribution to the field! It will provide an excellent springboard for future engagement with clerical vestments and the cult of Becket. I loved it.
I only have a questions/comments for the author to consider and I also suggest that the article be copyedited thoroughly, particularly in the footnotes and captions.
1. Introduction - was it normal to display a newly canonized saint in liturgical vestments, or was this unique to Becket for some reason?
Might be good in a note to observe, too, that the vestments should be examined in light of liturgical offices for the cult of the saint. See Sherry Reames ed. Liturgical Offices for the Cult of St. Thomas Becket (2000). Wonder if Katherine Emery, "Architecture, Space and Memory" (2020) would be a useful source for the liturgy and iconography of Becket?
2. Lines 139-143. Could you explain this more/unclear? Is this about the 2 swords doctrine? Eucharistic Controversy?
3. Lines 157-167 could use more clarification - why does having these Nicholas relics in Bari equate with Becket at Anagni? Here you could add Claudia Quattrocchi's recent article in Arts (2021).
4. Section around line 239. Is this a liturgical "kiss of peace"? Also, does the two-swords doctrine have the same meaning in the 13th and 14th centuries? Is papal power at issue in the same way it was in Becket's era?
5. around line 260 - if we are comparing these scenes to Romanesque architectural iconography (which is very interesting!), why are we not looking for sources in manuscripts? Also, if we are comparing the vestments that adorn the priestly body and architectural decoration, what can we say about the body of the church? the Body of Christ?
6. line 331 - interesting that a textile (pallium) is described in a textile?
7. line 398 - this issue of a ferula comes up frequently, particularly with Edward Grim. Was he a deacon? I lost track of his significance, but I wonder if the iconography with the ferula has more to it?
8. Line 466 - Did the interest in portraying this on liturgical vestments correlate to any waning of episcopal or papal power? Why this continuous interest in Becket and spiritual authority over time on vestments? Different between 12th c. politics and 14th?
9. Line 512 - who was Reginald Fitzurse?
Some editing suggestions:
Abstract: Should Liturgical Vestments be lowercase? Period at end of that sentence. Is "globally" an accurate term for the spread of the Becket cult in the "Western world"? Suggest transregional exchange? Lowercase "Patrons", and remove a period at the end of the abstract.
Note 2 - "in which the author does not specialize"; could we have a quick definition here as well for opus anglicanum?
Lines 54-65 - could we have a note that briefly defines the liturgical vestment terminology for the average reader? Not everyone knows what an orphrey or a cope is. Maybe a suggestion of a source with all of these terms define would be good?
Line 66 - we need an intro sentence to this paragraph on mitres. What will we learn in this section about these 4 mitres? What purpose will this section serve in terms of an overall thesis? (A quick sentence that mentions something from lines 168-169 would help).
I suggest more signposts in each section to guide the viewer to the conclusion section. Otherwise the whole article reads like a catalogue without a thesis. When I read the topic sentence, "The four mitres in the corpus form the oldest group of items," I wonder - so what?
line 73 previously *space* (Geijer)
Fig 2 - line 85 - remove a comma and add a period
line 98 need a space before 1170
line 147 St Lawrence (check saint vs St throughout)
line 173 and throughout - should Episcopal be uppercase?
line 177 - write out name of "M. Greenblatt" as you write out first name of authors elsewhere
Line 190 - same issue as mitres section - need a transition and a topic sentence that leads the viewer. Based on the topic sentence here, one would think this section is about how the corpus has remarkable examples. So what? While we are in this section, it is littered with the passive voice. Fix "is sat on a finely crafted seat" - awkward. Line 224 and 228 lowercase King.
Check spelling throughout for Abegg-Stiftung (see line 192)
line 203 p. 283 (add space)
line 359 - should pope be lowercase here?
line 616 - to which we will return.
line 660 - note 39 no italics.
Lots of minor mistakes in footnotes, including spaces before ":"; ".,", "(ed." with accents (line 949-950), no page numbers in line 958; British misspelled in line 962; sometimes pages number ranges need checking - line 923 vs 930.
Author Response
Thank you for all the comments and compliments you have made on my article.
1 -Thanks for the references on the relationship between Thomas Becket and the liturgy. I will include them in a note.
2 - Not at all. I am talking about the swords held by Henry II's knights (= bloody weapons) which contrast with the instruments of peace (= peace weapons) used by the servant of God, Becket. I will work on making the paragraph clearer.
3 - OK
4 - It is a ritualised kiss of peace. I am going to add this reference in footnote (Petkov, The Kiss of Peace: Ritual, Self, and Society 2003, pp. 64-5) and develop the section.
5 - Ackermann has already tried, and I indicate the relationships between the iconography of this object and the field of manuscripts. The Romanesque-inspired architectural setting indicates, in my opinion, that this cycle was probably inspired by older cycles that have now been lost, but which existed either in the textile field (altar fronts) and/or in the field of miniatures, or also in the monumental decoration. As you suggest it, the symbolic parallel in these embroidered panels may be numerous.
6. Absolutely. A setting in abyme.
7. Grim was a monk. But you are right that he is sometimes portrayed as a deacon in Becket's iconography (although sometimes tonsured).
8- Although the political context changed between the 12th and 14th centuries, Thomas Becket remained a particularly strong model of sanctity for the papal and episcopal milieu, which probably explains his presence in this embroidered production.
9- One of Henri II's knight
Thank you for all your editing suggestions that I will include in the article.
With my best regards.